# Compositional Scene Understanding through Inverse Generative Modeling

Yanbo Wang [1]   Justin Dauwels [1]   Yilun Du [2]

## Abstract

Generative models have demonstrated remarkable abilities in generating high-fidelity visual content. In this work, we explore how generative models can further be used not only to synthesize visual content but also to understand the properties of a scene given a natural image. We formulate scene understanding as an *inverse generative modeling* problem, where we seek to find conditional parameters of a visual generative model to best fit a given natural image. To enable this procedure to infer scene structure from images substantially different than those seen during training, we further propose to build this visual generative model *compositionally* from smaller models over pieces of a scene. We illustrate how this procedure enables us to infer the set of objects in a scene, enabling robust generalization to new test scenes with an increased number of objects of new shapes. We further illustrate how this enables us to infer global scene factors, likewise enabling robust generalization to new scenes. Finally, we illustrate how this approach can be directly applied to existing pretrained text-to-image generative models for zero-shot multi-object perception. Code and visualizations are at https://energy-based-model.github.io/compositional-inference.

## 1 Introduction

*"What I cannot create, I do not understand."*

– Richard Feynman

To understand surrounding physical scenes, human intelligence is able to learn abstract visual concepts from the physical world and compositionally reuse them (Biederman, 1987; Greff et al., 2020; Fodor & Lepore, 2002). Given an image of an object, we can then easily imagine how the object would look if it were rotated or moved in 3D

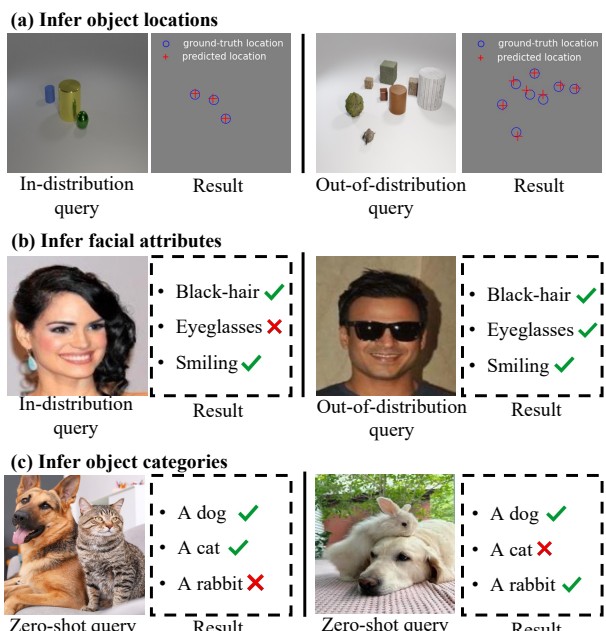

Figure 1: **Compositional Scene Understanding.** Our approach demonstrates strong generalization across various scene understanding tasks. For object location inference (first row), the model is trained on CLEVR images containing 3-5 objects, while the test set is CLEVRTex, which contains 6-8 objects. For multi-facial attribute inference (second row), the model is trained only on female faces from CelebA, and is tested exlusively on male faces. For object category inference (third row), we use pretrained Stable Diffusion without any additional fine-tuning, and the test set consists of multi-object natural images.

world (Shepard & Metzler, 1971). Such a generative learning mechanism is the key for us to accurately parse scenes that we have never encountered before, i.e., zero-shot scene understanding (Chomsky, 1965; Fodor & Pylyshyn, 1988; Bengio, 2019). We are interested in equipping machines with such generalizable scene-understanding abilities by leveraging recent advances in generative models.

Conventionally, scene understanding tasks have been dominated by discriminative models that learn a direct mapping from input images to visual attributes (Vapnik et al., 1998; Krizhevsky et al., 2012; Redmon, 2016), which, however, is demonstrated to struggle with generalizing to even slightly shifted test distributions (Geirhos et al., 2018; Recht et al., 2019; Taori et al., 2020; Hendrycks & Gimpel, 2016; Geirhos et al., 2020). In contrast, generative models have

---

[1]TU Delft [2]Harvard University. Correspondence to: Yanbo Wang <y.wang-27@tudelft.nl>.

*Proceedings of the 42nd International Conference on Machine Learning*, Vancouver, Canada. PMLR 267, 2025. Copyright 2025 by the author(s).

long been advocated for solving inference problems with the promise of better generalization brought by data generation modeling (Ng & Jordan, 2001; Hinton, 2007). Yet, only very recently have generative models begun to show promising results for visual inference tasks (Li et al., 2023a; 2024; Clark & Jaini, 2024), thanks to the highly expressive modeling abilities of diffusion models (Sohl-Dickstein et al., 2015; Ho et al., 2020). Despite this progress, these newly proposed generative inference approaches focus only on single-label classification tasks, and how to perform a broader range of scene understanding tasks (e.g., object discovery or multi-object classification) on scenes significantly more complex than those seen during training remains elusive.

In this work, we propose an *inverse generative modeling* framework that is broadly applicable across various scene understanding tasks, including those involving scenes more complex than that encountered during training. Our framework builds a visual generative model compositionally (Du & Kaelbling, 2024) from smaller generative pieces representing individual parts of a scene. During inference, to understand a scene, we aim to find the conditional parameters for a composed set of generative models that best fit a given natural image, enabling to fit more complex scenes by fitting a larger set of conditional parameters for more generative models.

In Fig. 1, we show how our approach can be used to compositionally interpret scenes across different visual understanding tasks. In the top row of Fig. 1, we illustrate how our approach can discover objects in a scene by predicting object positions and generalize effectivly to out-of-distribution images. For this task, the model is trained on *CLEVR* dataset with each image containing *3-5* objects, while tested on a different dataset *CLEVRTex* with *6-8* objects. The substantial difference in object number, shape, color, texture and background between the training set and the test set demonstrates the strong generalization ability of our approach. In the middle row of Fig. 1, we demonstrate how our approach can simultaneously classify multiple facial attributes on *CelebA* dataset and likewise generalize faithfully, where the training set contains only female faces while the test set contains only male faces. Finally, in the bottom row of Fig. 1, we show how our approach can adopt pretrained diffusion models to perform zero-shot multi-object perception task on web images without any additional training.

Our contributions are as follows: **(1)** We propose a generic inverse generative modeling framework to tackle several scene understanding tasks such as object discovery and zero-shot perception. **(2)** We build the inverse generative model compositionally, enabling strong generalization beyond training set. **(3)** Our approach significantly outperforms generative classifier baselines for several scene understanding tasks on both synthetic and realistic image datasets.

## 2 Related Work

**Generative Models for Visual Understanding.** Recent work has explored applying generative models to tasks beyond visual generation, such as classification (Li et al., 2023a; 2024; Jaini et al., 2023; Clark & Jaini, 2024; Mahajan et al., 2024; Chen et al., 2024), personalization (Gal et al., 2022; 2023; Avrahami et al., 2023), and segmentation (Amit et al., 2021; Brempong et al., 2022; Zhao et al., 2023; Wang et al., 2024). Most relevant to our work, generative classifiers (Li et al., 2024) leverage generative models to tackle single-label classification tasks. In contrast, our framework does not limit itself to solving single-label classification problems; instead, it demonstrates how generative models with flexible conditioning can address a broader range of visual understanding tasks, such as object discovery and zero-shot multi-object perception. More importantly, our approach composes a generative model from smaller submodels each capturing a specific visual concept, enabling generalizing to unseen scenes that differ substantially from training set.

**Compositional Generative Models.** There has been significant recent progress in incorporating compositionality into generative models (Du & Kaelbling, 2024) to enable generalization beyond training distribution (Du & Mordatch, 2019; Cho et al., 2023; Shi et al., 2023; Sohn et al., 2023; Du et al., 2020; 2021; 2023; Nie et al., 2021; Feng et al., 2022; Li et al., 2022; Liu et al., 2021; 2022; Huang et al., 2023; Cong et al., 2023; Wang et al., 2023; Su et al., 2024; Zhou et al., 2024; Netanyahu et al., 2024). While most of these works focus on generating novel scenes, we focus on a less explored direction – inverse compositional generative modeling for scene understanding. The most similar work in this direction is UCCD (Liu et al., 2023), which requires a group of images as input to identify common concepts across image clusters. In contrast, our approach takes a single image as input, aiming to discover visual concepts that best interpret it. Furthermore, unlike UCCD relying on text-to-image generative models, our approach leverages generative models with flexible conditioning and can be applied to a wider range of visual understanding tasks.

**Image Captioning.** Our work is also related to image captioning. By leveraging pre-trained text-to-image generative models (e.g., Stable Diffusion), our model can be applied to image captioning tasks like BLIP-2 (Li et al., 2023b). However, our approach is applicable to a broader range of scene understanding tasks beyond image captioning. For example, by conditioning on object coordinates, our approach can perform object discovery tasks and even enable generalization to more complex scenes (many more objects) than seen at training. This flexibility and generalizability distinguishes our approach from traditional image captioning models.

# 3 Compositional Scene Understanding through Inverse Generative Modeling

In this section, we introduce our inverse generative modeling approach for scene understanding tasks. Given an image $\boldsymbol{x}$, we aim to infer a set of $K$ visual components $\{\boldsymbol{c}^1, \cdots, \boldsymbol{c}^K\}$ that describe the image, where image $\boldsymbol{x}$ will often contain a larger or more complex combinations of concepts than those seen at training time. We first illustrate how we can model more complex test scenes by modeling the data generation process as a composition of a set of generative models. Next, we formulate how we can invert the generation process to infer the set of concepts that describe a given image.

## 3.1 Compositional Generative Modeling

In a visual domain, given a set of conditioned concepts $\{\boldsymbol{c}^1, \boldsymbol{c}^2, \cdots, \boldsymbol{c}^K\}$, we aim to construct a generative model that can accurately represent the probability distribution

$$p(\boldsymbol{x}|\boldsymbol{c}^1, \boldsymbol{c}^2, \ldots, \boldsymbol{c}^K), \tag{1}$$

over the space of images $\boldsymbol{x}$. The set of scenes with concepts $\{\boldsymbol{c}^1, \boldsymbol{c}^2, \cdots, \boldsymbol{c}^K\}$ can be much more complex at test time than those seen at training time, making it difficult to directly fit a generative model on the data.

One approach to model $p(\boldsymbol{x}|\boldsymbol{c}^1, \boldsymbol{c}^2, \ldots, \boldsymbol{c}^K)$ is to factorize the probability distribution (Du & Kaelbling, 2024) and approximate it as a product of simpler conditional distributions $p(\boldsymbol{x}|\boldsymbol{c}^k)$:

$$p(\boldsymbol{x}|\boldsymbol{c}^1, \ldots, \boldsymbol{c}^K) \propto \prod_{k=1}^{K} p(\boldsymbol{x}|\boldsymbol{c}^k). \tag{2}$$

While this approach is a biased approximation of $p(\boldsymbol{x}|\boldsymbol{c}^1, \ldots, \boldsymbol{c}^K)$, prior work (Liu et al., 2022) has found that it enables effective compositional generalization to a larger number of visual concepts. We can model each $p(\boldsymbol{x}|\boldsymbol{c}^k)$ as an energy-based model (EBM) $e^{-E_\theta(\boldsymbol{x}|\boldsymbol{c}^k)}$. The product distribution $p(\boldsymbol{x}|\boldsymbol{c}^1, \ldots, \boldsymbol{c}^K)$ takes the form of a summation of a set of energy functions:

$$p(\boldsymbol{x}|\boldsymbol{c}^1, \ldots, \boldsymbol{c}^K) \propto e^{-\sum_{k=1}^{K} E_\theta(\boldsymbol{x}|\boldsymbol{c}^k)}. \tag{3}$$

To parameterize this product of EBMs, similar to (Liu et al., 2022), we can represent each EBM $E_\theta(\boldsymbol{x}|\boldsymbol{c}^k)$ using the denoising function in diffusion model $\epsilon_\theta(\boldsymbol{x}^t|\boldsymbol{c}^k, t)$ which approximately represents $\nabla_{\boldsymbol{x}} E_\theta(\boldsymbol{x}|\boldsymbol{c}^k)$. To sample from the product distribution in Eqn (3) we can construct the composed denoising function:

$$\epsilon_\theta^{\text{comb}}(\boldsymbol{x}^t, t) = \sum_{k=1}^{K} \epsilon_\theta(\boldsymbol{x}^t, t|\boldsymbol{c}^k), \tag{4}$$

which approximately corresponds to $\nabla_{\boldsymbol{x}} \sum_{k=1}^{K} E_\theta(\boldsymbol{x}|\boldsymbol{c}^k)$. We can then use the composed denoising function $\epsilon_\theta^{\text{comb}}(\boldsymbol{x}^t, t)$ in the standard diffusion sampling process to approximately sample from the product distribution in Eqn (3).

To construct the composed noise prediction model in Eqn (4), prior work has focused on learning each denoising function $\epsilon_\theta(\boldsymbol{x}^t, t|\boldsymbol{c}^k)$ in isolation, combining denoising functions at test-time dependent on the composition needed. However, such a test-time composition of denoising functions can lead to the accumulation of error between score functions. To more accurately model the composed score function $\epsilon_\theta^{\text{comb}}(\boldsymbol{x}^t, t)$, we directly train the composed score function with the denoising diffusion objective:

$$\begin{aligned} \mathcal{L}_\theta &= \mathbb{E}_{\boldsymbol{x}, \epsilon, t} \|\epsilon - \epsilon_\theta^{\text{comb}}(\boldsymbol{x}^t, t)\|^2 \\ &= \mathbb{E}_{\boldsymbol{x}, \epsilon, t} \|\epsilon - \sum_{k=1}^{K} \epsilon_\theta(\boldsymbol{x}^t, t|\boldsymbol{c}^k)\|^2, \end{aligned} \tag{5}$$

where each of individual term $\epsilon_\theta(\boldsymbol{x}^t, t|\boldsymbol{c}^k)$ is parameterized by a neural network. This enables the composed denoising functions to behave more accurately together, and at test time, we can still compose additional terms of $\epsilon_\theta(\boldsymbol{x}^t, t|\boldsymbol{c}^k)$ to construct more complex scenes. We provide an overview of this training approach in Algorithm 1.

## 3.2 Compositional Scene Understanding

Given a generative model $p(\boldsymbol{x}|\boldsymbol{c}^1, \boldsymbol{c}^2, \ldots, \boldsymbol{c}^K)$, we can then formulate scene understanding given an image $\boldsymbol{x}$ as an inverse problem of finding parameters of the model that explain the image. Concretely, we seek to find a set of visual concepts $\hat{\boldsymbol{c}}^1, \ldots, \hat{\boldsymbol{c}}^K$ that maximize the log-likelihood of the observed image:

$$\hat{\boldsymbol{c}}^1, \ldots, \hat{\boldsymbol{c}}^K = \underset{\boldsymbol{c}^1, \ldots, \boldsymbol{c}^K}{\arg\max} \log p(\boldsymbol{x}|\boldsymbol{c}^1, \boldsymbol{c}^2, \ldots, \boldsymbol{c}^K). \tag{6}$$

The inferred set of concepts $\boldsymbol{c}^k$ then corresponds to a description of the scene, where individual concepts can flexibly describe individual objects of the scene as well as global features.

We can approximate the optimization of likelihood in Eqn (6) in diffusion models through the variational lower bound. The variational bound corresponds to a weighted form of the objective:

$$\hat{\boldsymbol{c}}^1, \ldots, \hat{\boldsymbol{c}}^K = \underset{\boldsymbol{c}^1, \ldots, \boldsymbol{c}^K}{\arg\min} \mathbb{E}_{\epsilon, t} \|\epsilon - \epsilon_\theta(\boldsymbol{x}^t, t|\boldsymbol{c}^1, \ldots, \boldsymbol{c}^K)\|^2,$$

where similar to prior work, we can approximate the likelihood by ignoring the weighting terms on each loss (Li et al., 2023a). Thus, for a given image, we can optimize for a set of visual concepts by minimizing the above objective.

We can then use the approach discussed in Section 3.1 to directly parameterize denoising functions for more complex scenes with more concepts by optimizing Eqn (4), leading to the optimization objective of:

$$\hat{\boldsymbol{c}}^1, \ldots, \hat{\boldsymbol{c}}^K = \underset{\boldsymbol{c}^1, \ldots, \boldsymbol{c}^K}{\arg\min} \mathbb{E}_{\epsilon, t} \|\epsilon - \sum_{k=1}^{K} \epsilon_\theta(\boldsymbol{x}^t, t|\boldsymbol{c}^k)\|^2, \tag{7}$$

where we use a set of $N$ samples with different sampled

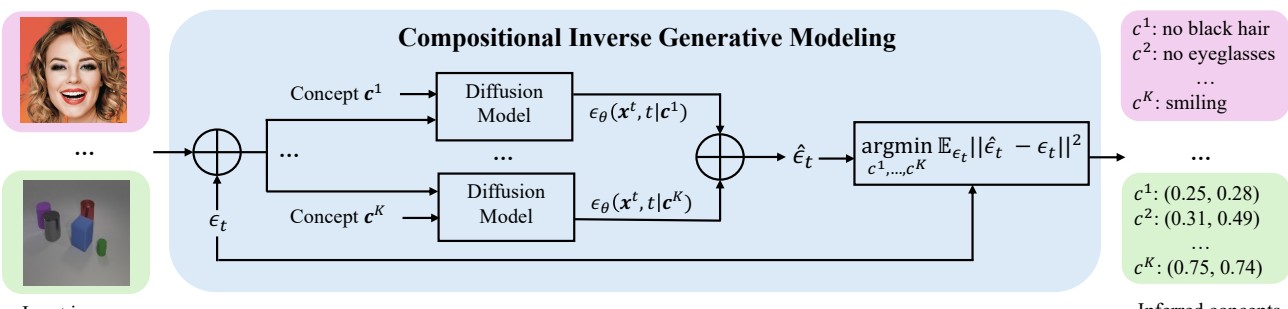

Figure 2: **Compositional Scene Understanding**. Our model achieves scene understanding by identifying the optimal conditioning concepts (e.g., facial attributes or object coordinates) that best interpret the input test image. Its compositional structure allows for simultaneous inference of multiple concepts and enables robust generalization to images that differ substantially from the training data.

---

**Algorithm 1** Training Algorithm

1: **Input:** data distribution $p_D$, denoising model $\epsilon_\theta$
2: **while** not converged **do**
3:    $(\boldsymbol{x}_0, \boldsymbol{c}^1, \boldsymbol{c}^2, ..., \boldsymbol{c}^K) \sim p_D$
4:    ▷ *Compute denoising direction*
5:    $\epsilon \sim \mathcal{N}(0, 1), t \sim \text{Unif}(\{1, \ldots, T\})$
6:    $\boldsymbol{x}^t = \sqrt{\bar{\alpha}_t}\boldsymbol{x}_0 + \sqrt{1 - \bar{\alpha}_t}\epsilon$
7:    $\Delta\theta \leftarrow \nabla_\theta \|\epsilon - \sum_{k=1}^K \epsilon_\theta(\boldsymbol{x}^t, t, \boldsymbol{c}^k)\|^2$
8: **end while**
9: **return** $\epsilon_\theta$

---

timesteps and noise to estimate the objective.

**Optimizing Visual Concepts.** We can solve Eqn (7) with different optimization algorithms, depending on the concepts $\boldsymbol{c}^k$ are discrete or continuous in specific visual understanding tasks. When each visual concept $\boldsymbol{c}^k \in \{\ell_1^k, \ell_2^k, ..., \ell_M^k\}$ is a discrete variable with a finite set of possibilities, we can directly optimize Eqn (7) by enumerating through each possible configuration of $\boldsymbol{c}^k$ and evaluating the average denoising error. We illustrate this optimization in Algorithm 2. To scale to a large number of discrete concepts and reduce inference time, we further propose a gradient-based search method in Algorithm 5 in Sec A.2. In contrast, when $\boldsymbol{c}^k$ is continuous, such as when they describe the locations of objects in the scene, optimization is substantially more complex, as exhaustive search is not feasible and gradient-based optimization is easily susceptible to local minima. We describe more complex algorithms for inference when dealing with this setting in Section 3.3.

**Inferring Number of Visual Concepts.** In many scene understanding tasks, it is difficult to know beforehand the number of visual concepts $K$ in the scene. For instance, in an object discovery task, the number of concepts (objects) may differ from one image to another. To determine the number of concepts $K$ for a given test scene before inferring concept parameters, we can find a number $\hat{K}$ that maximize the log-likelihood of the test image:

$$\hat{K} = \underset{K \in [K_{min}, K_{max}]}{\arg\max} \left\{ \max_{\boldsymbol{c}^1, ..., \boldsymbol{c}^K} \log p(\boldsymbol{x}|\boldsymbol{c}^1, \ldots, \boldsymbol{c}^K) \right\}, \quad (8)$$

---

**Algorithm 2** Discrete Concept Inference Algorithm

1: **Require:** an image $\boldsymbol{x}$, trained denoising model $\epsilon_\theta$
2: Determine all possible $M^K$ concept configurations $\boldsymbol{c}_{tuple} = \{(\boldsymbol{c}^1, \boldsymbol{c}^2, ..., \boldsymbol{c}^K)|\boldsymbol{c}^k \in \{\ell_1^k, \ell_2^k, ..., \ell_M^k\}\}$
3: ▷ *Evaluate denoising error for each configuration*
4: Initialize a denoising error list $\mathbf{E} = \text{zeros}(M^K)$
5: **for** $n = 1, \ldots, N_{\text{sample}}$ **do**
6:    $\epsilon_n \sim \mathcal{N}(0, 1), t_n \sim \text{Unif}(\{1, \ldots, T\})$
7:    $\boldsymbol{x}^{t_n} = \sqrt{\bar{\alpha}_t}\boldsymbol{x} + \sqrt{1 - \bar{\alpha}_t}\epsilon_n$
8:    **for** $j = 1, \ldots, M^K$ **do**
9:      $\mathbf{E}[j]$ += $\|\epsilon_n - \sum_{k=1}^K \epsilon_\theta(\boldsymbol{x}^{t_n}, t_n, \boldsymbol{c}_{tuple}^k[j])\|^2$
10:    **end for**
11: **end for**
12: ▷ *Select the configuration with lowest denoising error*
13: $\hat{j} = \arg\min_{j \in \{1, ..., M^K\}} \frac{1}{N}\mathbf{E}[j]$
14: **return** $\boldsymbol{c}_{tuple}[\hat{j}]$

---

where $K_{min}$ and $K_{max}$ are the minimal and maximal limit of $K$. Similar to Eqn (6)-Eqn (7), we can approximate the log-likelihood in Eqn (8) with variational lower bound and parameterize the denoising function with a composition of multiple functions for generalization purpose:

$$\hat{K} = \underset{K \in [K_{min}, K_{max}]}{\arg\min} \left\{ \min_{\boldsymbol{c}^1, ..., \boldsymbol{c}^K} \mathbb{E}_{\epsilon, t} \|\epsilon - \sum_{k=1}^K \epsilon_\theta(\boldsymbol{x}^t, t|\boldsymbol{c}^k)\|^2 \right\}.$$

In Figure 3, we visually illustrate how $\hat{K}$ can be determined in object discovery tasks by solving the above expression. We can see that the ground truth number consistently yields the lowest average denoising error (highest likelihood), demonstrating the effectiveness of our concept number determination approach. We illustrate the object location visualization in Figure XII and the algorithm for determining the number of concepts in Algorithm 4 in the Appendix.

Overall, our proposed inverse generative modeling (IGM) framework significantly broadens the applicability of generative models to visual understanding tasks with several key elements: (1) flexible conditioning enables the inference of continuous concepts beyond class labels; (2) compositional modeling supports simultaneous multi-concept inference

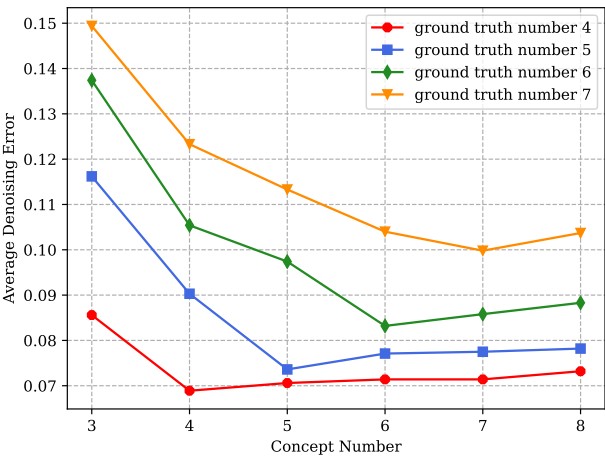

Figure 3: **Concept Number Inference.** Illustration of object number inference on CLEVR. Given a test image, our model evaluates each number $K \in \{3, .., 8\}$ respectively by using $K$ objects to fit the image and obtain corresponding denoising errors. Out of the potential options $K \in \{3, ..., 8\}$, our model determines the one with the lowest denoising error as object number, which turns out to be consistent with the ground truth number.

besides single-concept inference; (3) compositionality allows generalization to scenes substantially different from training data; and (4) the inference algorithm is applicable to both domain-trained diffusion models and generic pretrained diffusion models. An overview of our proposed inverse generative modeling approach is illustrated in Figure 2. In Section 4.3, we demonstrate how our model can adopt pretrained text-to-image generative models like Stable Diffusion to solve zero-shot multi-object perception tasks without requiring any additional training.

### 3.3 Continuous Visual Concept Inference

In Section 3.2, we aim to infer a set of concepts that best describe a given image by optimizing Eqn (7). When concepts $c^k$ are continuous, however, the optimization using gradient descent faces several practical challenges. First, the potential non-convexity of the objective function, due to the neural network parametrization of $\epsilon_\theta(\boldsymbol{x}^t, t | c^k)$, can cause $c^k$ to converge to local minima, resulting in substantial deviation from the optimal solution. Second, evaluating the expectation term in Eqn (7) at every gradient descent step incurs prohibitively high sample complexity with respect to $\epsilon_n$ and $t_n$, as well as significant computational complexity for evaluating $\epsilon_\theta(\boldsymbol{x}^t, t | c^k)$. We propose improved strategies on top of gradient descent to overcome these challenges as illustrated below and outlined in Algorithm 3.

**Effective Concept Initialization.** To more effectively prevent $c^k$ from converging to local minima, we propose initializing $c^k$ with multiple random starting points, denoted as $c_1^k, c_2^k, ..., c_R^k$, and maintaining corresponding updates throughout the optimization process. After every few optimization steps, we terminate paths with low log-likelihood

---

**Algorithm 3** Continuous Concept Inference Algorithm

1: **Require:** an image $\boldsymbol{x}$, trained denoising model $\epsilon_\theta$, $\lambda$
2: ▷ *Initialize multiple (R) sets of concepts*
3: Initialize concepts $\{c_r^1, c_r^2, ..., c_r^K\}_{r=1}^R \sim \mathcal{N}(0, 1)$
4: ▷ *Run Stochastic Gradient Descent*
5: **for** $n = 1, \ldots, N_{\text{step}}$ **do**
6: $\quad \epsilon_n \sim \mathcal{N}(0, 1), t_n \sim \text{Unif}(\{1, \ldots, T\})$
7: $\quad \boldsymbol{x}^{t_n} = \sqrt{\bar{\alpha_{t_n}}} \boldsymbol{x} + \sqrt{1 - \bar{\alpha_{t_n}}} \epsilon_n$
8: $\quad \Delta c_r^k \leftarrow \nabla_{c_r^k} \|\epsilon_n - \sum_{k=1}^K \epsilon_\theta(\boldsymbol{x}^{t_n}, t_n, c_r^k)\|^2$
9: $\quad c_r^k \leftarrow c_r^k - \lambda \Delta c_r^k$
10: **end for**
11: ▷ *Evaluate denoising error for each set*
12: Initialize a denoising error list $\mathbf{E} = \text{zeros}(R)$
13: **for** $i = 1, \ldots, N_{\text{sample}}$ **do**
14: $\quad \epsilon_i \sim \mathcal{N}(0, 1), t_i \sim \text{Unif}(\{1, \ldots, T\})$
15: $\quad \boldsymbol{x}^{t_i} = \sqrt{\bar{\alpha_{t_i}}} \boldsymbol{x} + \sqrt{1 - \bar{\alpha_{t_i}}} \epsilon_i$
16: $\quad$ **for** $r = 1, \ldots, R$ **do**
17: $\quad\quad \mathbf{E}[r] \mathrel{+}= \|\epsilon_i - \sum_{k=1}^K \epsilon_\theta(\boldsymbol{x}^{t_i}, t_i, c_r^k)\|^2$
18: $\quad$ **end for**
19: **end for**
20: ▷ *Select the set with lowest denoising error*
21: $\hat{r} = \text{argmin}_{r \in \{1, ..., R\}} \frac{1}{N} \mathbf{E}[r]$
22: **return** $c_{\hat{r}}^1, c_{\hat{r}}^2, ..., c_{\hat{r}}^K$

---

values, as the optimal $c^k$ is expected to yield a high likelihood. This process ultimately converges to a single optimal configuration of $c^k$ with the highest likelihood. Empirically, we find that this initialization strategy improves the algorithm's ability to escape local minima, thereby significantly enhancing scene understanding accuracy, as demonstrated in the ablation study in Sec 4.1 and Table IV.

**Efficient Concept Optimization.** To address the sample and computation complexity associated with evaluating the expectation term in Eqn (7), we propose leveraging stochastic gradient descent (SGD) for optimization. This approach requires a single sample of $\epsilon_n$ and $t_n$ at each optimization step to update the concepts $c^k$. As a result, the sample complexity is reduced from $N$ to 1 per iteration, and $\epsilon_\theta(\boldsymbol{x}^t, t | c^k)$ needs to be evaluated only once per iteration in stead of $N$ times. This significantly accelerates the inference speed.

## 4 Experiments

In this section, we evaluate the scene understanding capabilities of our proposed approach across three different tasks. First, we consider a local factor perception task in Section 4.1, where the objective is to infer the center coordinates of objects. We next perform a global factor perception task to predict facial attributes from human faces in Section 4.2. Finally, we demonstrate how our approach can be adapted to pretrained models for zero-shot multi-object perception without any additional training in Section 4.3.

| Models | In-distribution (3-5 objects) | | Out-of-distribution (6-8 objects) | |
|---|---|---|---|---|
| | Perception Rate ↑ | Estimation Error ↓ | Perception Rate ↑ | Estimation Error ↓ |
| ResNet-50 (He et al., 2016) | 5.3% | $19.4e^{-2}$ | 2.9% | $19.7e^{-2}$ |
| SlotAttn (Locatello et al., 2020) | 80.4% | $8.7e^{-4}$ | 53.3% | $1.3e^{-3}$ |
| DINOSAUR (Seitzer et al., 2022) | 82.5% | $8.4e^{-4}$ | 59.0% | $1.2e^{-3}$ |
| GC (Li et al., 2024) | 82.2% | $6.0e^{-4}$ | 58.7% | $1.2e^{-3}$ |
| IGM w/o multiple-initialization (Ours) | 72.8% | $6.9e^{-4}$ | 68.0% | $7.8e^{-4}$ |
| IGM with multiple-initialization (Ours) | **94.7%** | $\mathbf{1.4}e^{-4}$ | **85.3%** | $\mathbf{3.5}e^{-4}$ |

Table 1: **Accuracy of Object Discovery.** Quantitative evaluation of object perception results on CLEVR for both in-distribution (3-5 objects) and out-of-distribution (6-8 objects) test settings. Perception rate and estimation error are reported. Our approach outperforms all the baselines, and the margin is especially significant for the out-of-distribution setting, demonstrating strong generalization capability.

## 4.1 Local Factor Perception

We demonstrate how our approach can infer local factors, such as object coordinates, from a test image and effectively generalize to scenes containing a larger number of objects and more complex objects than those seen during training.

**Dataset.** We evaluate our approach on the CLEVR dataset (Johnson et al., 2017), where each image is annotated with ground truth center coordinates of the objects. The training set consists of images containing *3-5 objects*. To evaluate the generalization ability of our approach on out-of-distribution data, we consider two settings: (1) images from the CLEVR dataset containing *6-8 objects*; (2) images from the *CLEVRTex* dataset containing *6-8 objects*.

**Baselines.** We compare our approach against both discriminative and generative baselines, including ResNet-50 (He et al., 2016), Slot Attention (SlotAttn) (Locatello et al., 2020), DINOSAUR (Seitzer et al., 2022), and Generative Classifier (GC) (Li et al., 2024). Details on how these baselines are trained for the object perception task can be found in Appendix A.5.

**Metrics.** We evaluate the object discovery performance in terms of object perception rate and coordinate estimation error. The object perception rate measures the percentage of correctly discovered object relative to the total number of objects. To determine which objects are correctly discovered, we use Hungrian algorithm to match predicted coordinates and ground truth coordinates for all object in the scene. An object is considered successfully discovered if the mean square error (MSE) between the predicted coordinates and the ground truth coordinates is less than 0.002. The coordinate estimation error is computed by averaging the MSE of the predicted coordinates and the ground truth coordinates across all objects.

**Qualitative Results.** We qualitatively illustrate that our approach can infer object coordinates from test images more accurately than baseline models, as shown in Figure 4. Furthermore, we demonstrate how our approach can generalize to images with a larger number of objects and more complex objects in Figure 5. On the left of Figure 5, our model can

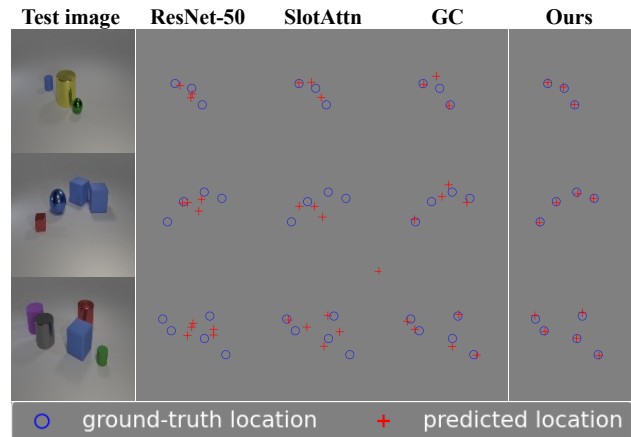

Figure 4: **In-distribution Object Discovery.** We train our model with CLEVR images containing 3-5 objects. During inference, given an in-distribution image (also containing 3-5 objects), our approach accurately identifies object coordinates. Compared with both determinative and generative baselines, our proposed approach demonstrates better coordinates estimation performance.

successfully generalize to images with 6-8 objects despite being trained only on images containing 3-5 objects. In contrast, all baseline models predict object locations that significantly deviate from the ground truth. On the right of Figure 5, we highlight the faithful generalization ability of our model in even more challenging scenarios, where the test images are from a different dataset CLEVRTex featuring substantially different colors, textures and backgrounds compared to the training set. In this setting, our approach can still predict object location accurately, while baselines predict random location. Additional qualitative results are provided in Figure VIII and Figure IX.

**Quantitative Results.** We quantitatively compare our approach with baselines in Table 1. Our method and Generative Classifier demonstrate better perception performance than the discriminative baselines ResNet-50 and Slot Attention, and our approach, by compositional modeling, achieves a 12.5% higher perception rate than Generative Classifier, with the margin increasing to 26.6% on the out-of-distribution tests. Meanwhile, our approach exhibits significantly lower coordinate estimation error than base-

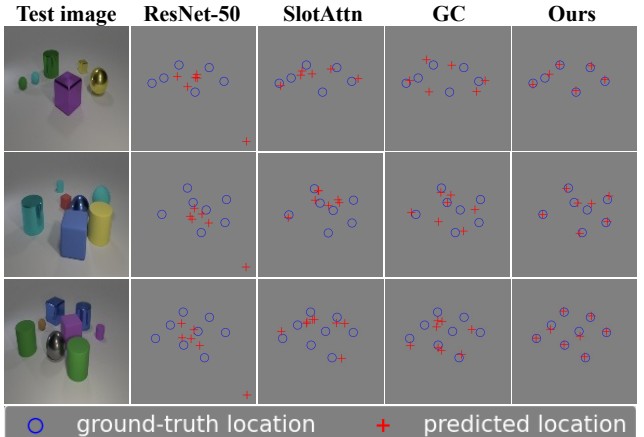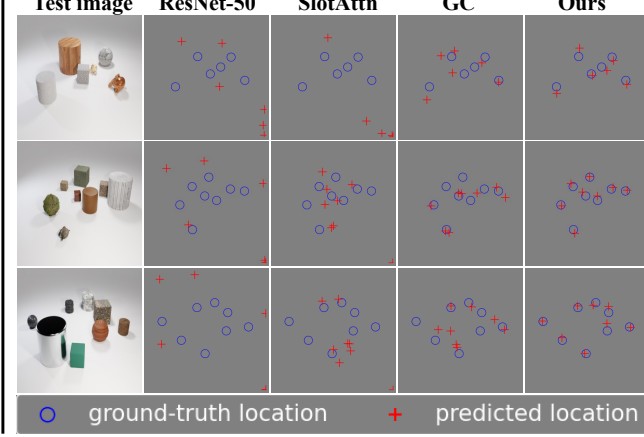

Figure 5: **Out-of-distribution Object Discovery.** Object perception results on out-of-distribution images: CLEVR images with 6-8 objects (**Left**) or CLEVRTex images with 6-8 objects (**Right**). Our model is trained with CLEVR images containing 3-5 objects. During inference time, given an out-of-distribution image that is substantially different from training data, our proposed approach can still infer the object positions accurately. In contrast, all baseline models predict object locations that significantly deviate from the ground truth.

lines, especially in out-of-distribution tests. Overall, these quantitative results demonstrate the strong generalization capability of our proposed approach to more complex scenes than those seen during training.

**Ablation Study.** We further demonstrate the effectiveness of our proposed random multiple-initialization strategy in Sec 3.3. As shown in Table 1, omitting the random multiple-initialization strategy leads to a significant degradation in object perception performance. In this case, the algorithm often converges to local minima that substantially deviate from the ground truth coordinates, even with an increased number of optimization steps. In contrast, adopting our proposed random multiple-initialization strategy significantly improves the perception performance. Additional ablation study results can be found in Table IV.

### 4.2 Global Factor Perception

We further illustrate how our approach can infer global factors, such as facial attributes, from a test image and reliably generalize to images that differ substantially from training data.

**Dataset.** We evaluate our approach on the CelebA dataset (Liu et al., 2015) focusing on three attributes: Black Hair, Eyeglasses, and Smiling. The training set consists only *female faces* labeled with the these attributes, while the out-of-distribution test set comprises solely *male faces*.

**Baselines.** We compare our approach with both discriminative and generative approaches including ResNet-50 (He et al., 2016), Generative Classifier (GC) (Li et al., 2024), and a variant of Generative Classifier. Details on how these baselines are trained for the facial feature perception task can be found in Appendix A.5 .

| Models | In-distribution (female faces) | Out-of-distribution (male faces) |
|---|---|---|
| ResNet-50 | 79.6% | 62.2% |
| GC (Li et al., 2024) | 79.1% | 61.7% |
| GC Variant | 77.8% | 58.1% |
| IGM (Ours) | **80.8%** | **65.6%** |

Table 2: **Accuracy of Facial Feature Prediction.** Quantitative evaluation of facial feature prediction results for both in-distribution (female faces) and out-of-distribution (male faces) settings on CelebA. Our model outperforms all baseline approaches in terms of classification accuracy for the in-distribution setting and generalize even much better for the out-of-distribution setting.

**Metrics.** We evaluate facial attribute prediction performance with classification accuracy. Classification accuracy is defined as the ratio of correctly classified images to the total number of images, where an image is considered correctly classified only if all the three attributes are simultaneously predicted correctly.

**Qualitative Results.** We demonstrate how our approach can predict the presence of all three facial attributes in a given face image in Figure 6. On the left of Figure 6, we show that, by explicitly composing the three attributes with compositional diffusion models, our approach provides more accurate facial attribute prediction results than baselines. On the right of Figure 6, we further illustrate how our approach can faithfully predict facial attributes even in male faces, despite never having seen that during training, demonstrating stronger generalization compared to baselines. Additional qualitative results are provided in Figure X.

**Quantitative Results.** We quantitatively compare our approach with baselines in Table 2. Our approach outperforms all baselines in classification accuracy for in-distribution images, and the performance gap becomes even pronounced for the out-of-distribution tests. This strong generalization

Figure 6: **In-Distribution and Out-of-Distribution Facial Feature Prediction.** Facial feature prediction results for in-distribution (**Left**) and out-of-distribution (**Right**) CelebA images. Our model is trained on female faces from CelebA. During inference, our model can accurately predict facial features consistent with the ground truth for both in-distribution female faces and out-of-distribution male faces.

to such non-trivial distribution shifts demonstrates the robustness of our compositional modeling strategy.

## 4.3 Zero-Shot Multi-Object Perception

Finally, we demonstrate that our approach can leverage pretrained diffusion models, such as Stable Diffusion (Rombach et al., 2022), for zero-shot multi-object perception tasks without requiring any additional training. Specifically, we compose a set of diffusion models, each conditioned on an individual text prompt, and minimize the average denoising error with respect to the text prompts following Eqn (7). The solution can then be obtained using Algorithm 2.

In our experiment, we evaluate our model on a small dataset as detailed in Section A.3, each containing two animals from the set {dog, cat, rabbit}. The prompts corresponding to these object concepts are: "a photo of dog", "a photo of cat", and "a photo of rabbit". Our model composes two diffusion models, each conditioned on any two of the prompts to interpret a given image; evaluates denoising error for the three possible prompt combinations; and selects the combination with the lowest denoising error as the optimal solution. In contrast, Diffusion Classifier (DC) (Li et al., 2023a) baseline uses a single diffusion model conditioned on compound prompts (e.g., "a photo of a dog and a cat") without explicitly modeling compositionality. The Diffusion Classifier Variant (DC variant) baseline also uses a single diffusion model but conditioned on individual object prompts, and select the two prompts with smallest denoising error as perception results. Details on Diffusion Classifier and the variant for multi-object perception can be found in Appendix A.5.

As shown in Figure 7 and Figure XI, our approach can consistently recognize multiple objects in realistic images using pretrained generative models without any training. We further quantitatively compare our approach with baselines in Table 3, where our approach outperforms Diffusion Clas-

Figure 7: **Zero-Shot Multi-Object Perception.** Our approach can faithfully interpret given real-world images by predicting object categories that are consistent with the ground truth.

| Models | Accuracy ↑ |
|---|---|
| Diffusion Classifer (Li et al., 2023a) | 70.4% |
| Diffusion Classifer Variant | 73.2% |
| IGM (Ours) | **87.3%** |

Table 3: **Accuracy of Zero-Shot Multi-Object Perception.** By adopting pretrained Stable Diffusion without any additional training, our compositional approach significantly outperforms Diffusion Classifier and its variant on real-world images in terms of classification accuracy for the multi-object perception task.

sifier by a margin of 16.9% in perception accuracy. This demonstrates that our proposed compositional generative modeling framework effectively enables multi-object scene understanding through leveraging pretrained models.

## 5 Limitations and Conclusion

**Limitations.** For multiple discrete concept inference, our compositional modeling approach enumerates through each possible configuration for every concept and evaluate denoising error for all concept combinations. This can result in long inference time when the concept number is large. To scale to a large number of concept settings, we

developed a continuous approximation of our approach that allows gradient-based optimization in Algorithm 5 and Algorithm 6, thereby avoiding the exponential inference cost. Alternatively, several additional approaches can potentially significantly mitigate this computational bottleneck. One approach could be to use heuristic search algorithms on discrete values – for instance we can run beam search with a beam width of $K$ over each attribute sequentially which can reduce time complexity from $O(M^K)$ to $O(MK)$, making inference more efficient for large discrete spaces. Finally, since our approach allows parallel processing across the configurations, inference time can be drastically reduced given sufficient computational resources, potentially approaching the time required for a single configuration evaluation.

Another limitation is the assumption of concept independence. Our compositional generative modeling approach assumes object concept independence given the input image, enabling combinatorial generalization beyond the training distribution. However, one possible limitation of this full independence approximation is that it ignores the interaction between objects, which are crucial in many real-world scenarios. As a remedy, we could potentially learn additional models that model interactions between object components, which can also be composed to represent more complex scenes.

**Conclusion.** We have presented an inverse generative modeling approach to scene understanding tasks by compositionally combining a set of generative models. We illustrate how compositionality enables the inference of visual concepts from test images that differ substantially from training data. By adopting pretrained text-to-image generative models, our model can even achieve zero-shot multi-object perception without requiring any additional training. We believe that exploring compositions of various foundation models at test time can be an promising direction to build intelligent perception systems that can generalize more effectively.

## Impact Statement

No immediate negative social impact is anticipated from our proposed approach in its current form, as we focus primarily on scene understanding tasks using standard dataset. Given the strong generalization capability to handle more complex scenes than those seen during training, our approach has the potential to benefit various fields such as autonomous driving, robot manipulation, and augmented reality, among others. Furthermore, our approach can be environmentally friendly, since our framework can leverage pretained text-to-image generative models directly for zero-shot multi-object scene understanding tasks without requiring any additional training, thereby reducing the carbon footprint.

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

# A  Appendix

In the Appendix, we first present additional results for various scene understanding tasks in Sec. A.1. Next, we conduct further experiments in Sec. A.2, and provide dataset details in Sec. A.3, training details in Sec. A.4, and baseline details in Sec. A.5. Finally, we discuss potential extensions of our approach in Sec. A.6.

## A.1  Additional Results

**Local Factor Perception.**    We illustrate additional qualitative results for the object discovery task in Figure VIII and Figure IX. Our model, despite trained only on CLEVR images with 3-5 objects, not only successfully generalizes to CLEVR images containing a larger number (6-8) of objects, but also to CLEVRTex images containing objects with substantially different colors, shapes, textures and backgrounds compared to the training images. These qualitative results demonstrate how our proposed compositional modeling by composing a set of diffusion models enables strong generalization beyond training data.

**Global Factor Perception.**    We further illustrate additional qualitative results for the facial attribute prediction task in Figure X. We train our model on female face images only from the CelebA dataset to predict facial attributes: Black Hair, Eyeglasses, and Smiling. During inference, our model is tested on male face images that differ significantly from the training images. As shown in Figure X, our model can accurately predict facial attributes from male faces, demonstrating strong ability to generalize to non-trival distribution shift.

**Zero-Shot Multi-Object Perception.**    We illustrate additional qualitative results for the zero-shot multi-object perception task in Figure XI. We apply our proposed compositional inverse generative modeling framework directly to pretrained Stable Diffusion with any further training. Given real-world images, our model can predict object categories accurately, demonstrating effective zero-shot scene understanding ability.

## A.2  Additional Experiments

**Visualization of Concept Number Inference.**    In Figure 3, we demonstrate how denoising error can serve as a criterion for selecting concept number in the object discovery task. To more intuitively motivate this approach, we further illustrate visualization results in Figure XII to show how inferred concepts differ from ground truth concepts when the concept number does not match the ground truth concept number. We can see that when the concept number differs from the ground truth concept number, either some concepts are missed or extra concepts are inferred. This concepts mismatch results in a large denoising error, and

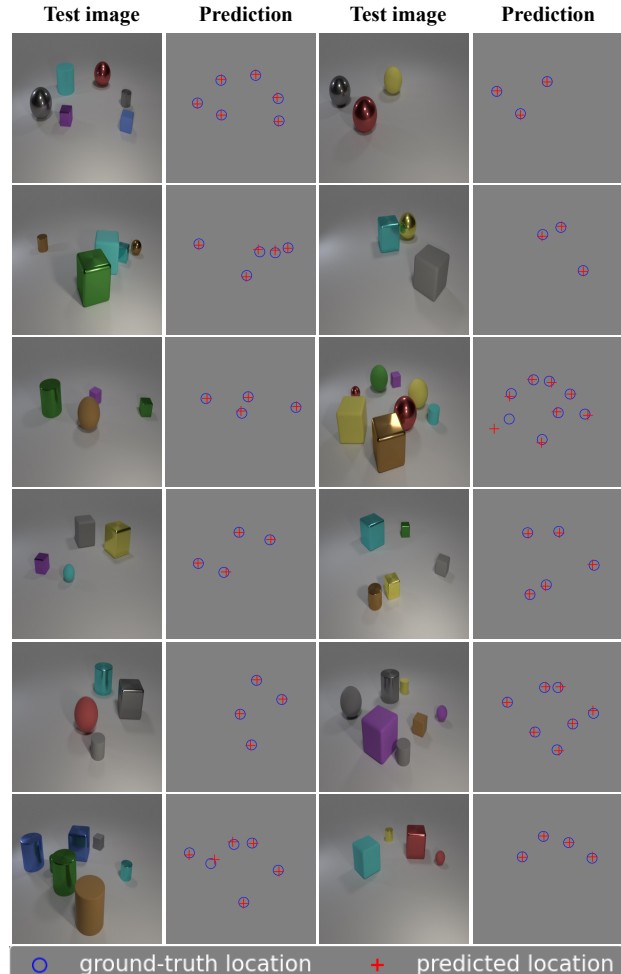

| Test image | Prediction | Test image | Prediction |
|---|---|---|---|

Figure VIII: **Object Discovery Generalization.** Object discovery results on CLEVR images containing 3-8 objects. Despite trained with CLEVR images containing 3-5 objects, our model can effectively generalize to scenes with a larger number of objects.

only the ground truth number can best fit a given image and leads to small denoising error, as reflected in Figure 3 and Figure XII. Similar to Algorithm 3, we outline the concept number inference algorithm in Algorithm 4, where we examine each possible concept number $K = K_{min}, ..., K_{max}$, figure out best concepts $c^k$ under each $K$, evaluate average denoising error for each $K$, and select the one configuration of $K$ with smallest average denoising error as the concept number estimate.

**Multiple Random Initialization Strategy Ablation.**    We discussed the importance of multiple random initialization strategy for continuous concept inference on top of Stochastic Gradient Descent (Amari, 1993) in Section 3.3 and we presented the results of an ablation study in Table 1. To further demonstrate the effectiveness of this approach, we visually illustrate how single random initialization may converge to either an optimal solution or a local minima in Fig-

| Models | In-distribution (3-5 objects) | | Out-of-distribution (6-8 objects) | |
|---|---|---|---|---|
| | Perception Rate ↑ | Estimation Error ↓ | Perception Rate ↑ | Estimation Error ↓ |
| IGM 1-initialization (Ours) | 72.8% | $6.9e^{-4}$ | 68.0% | $7.8e^{-4}$ |
| IGM 5-initialization (Ours) | 89.6% | $2.0e^{-4}$ | 79.1% | $5.4e^{-4}$ |
| IGM 10-initialization (Ours) | 90.5% | $1.9e^{-4}$ | 81.6% | $4.6e^{-4}$ |
| IGM 15-initialization (Ours) | 92.8% | $1.6e^{-4}$ | 84.3% | $3.5e^{-4}$ |
| IGM 20-initialization (Ours) | **94.7%** | **$1.4e^{-4}$** | **85.3%** | **$3.5e^{-4}$** |

Table IV: **Ablation Study of Multiple Random Initialization Strategy.** We illustrate a quantitative evaluation of object perception results on CLEVR for both in-distribution (3-5 objects) and out-of-distribution (6-8 objects) test settings. The object coordinates are inferred using Algorithm 3 with varying numbers of random initializations for all concepts. The quantitative results indicate that as the number of initialization starting points increases, the perception performance improves consistently. This demonstrates that our proposed multiple random initialization strategy can help avoid convergence to local minima for continuous concept inference.

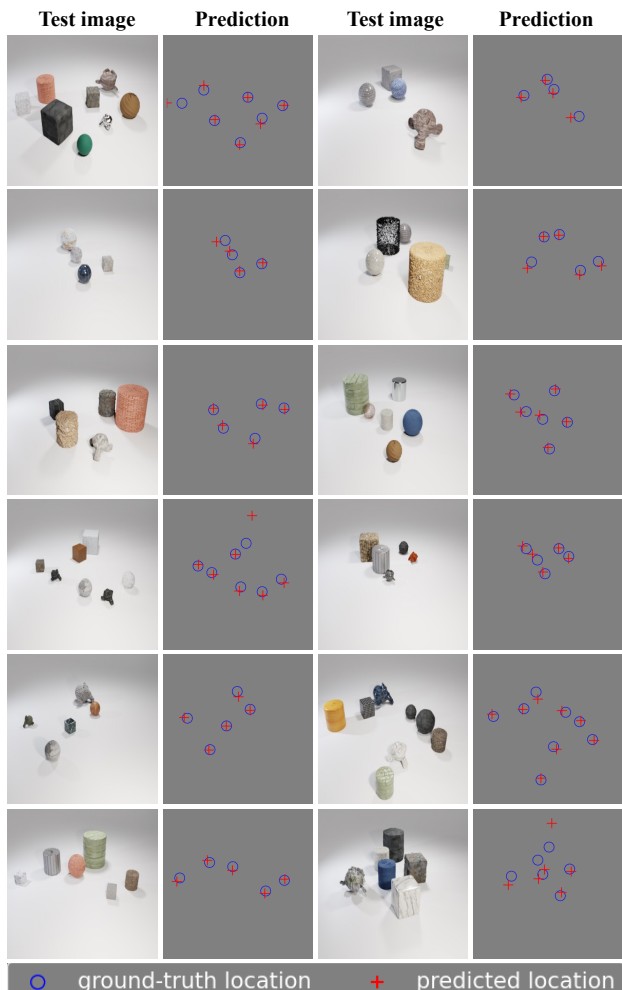

Figure IX: **Object Discovery Generalization.** Object discovery results on CLEVRTex images containing 3-8 objects. Despite trained with CLEVR images containing 3-5 objects, our model can effectively generalize to new CLEVRTex scenes containing a larger number of objects with different colors, shapes and textures.

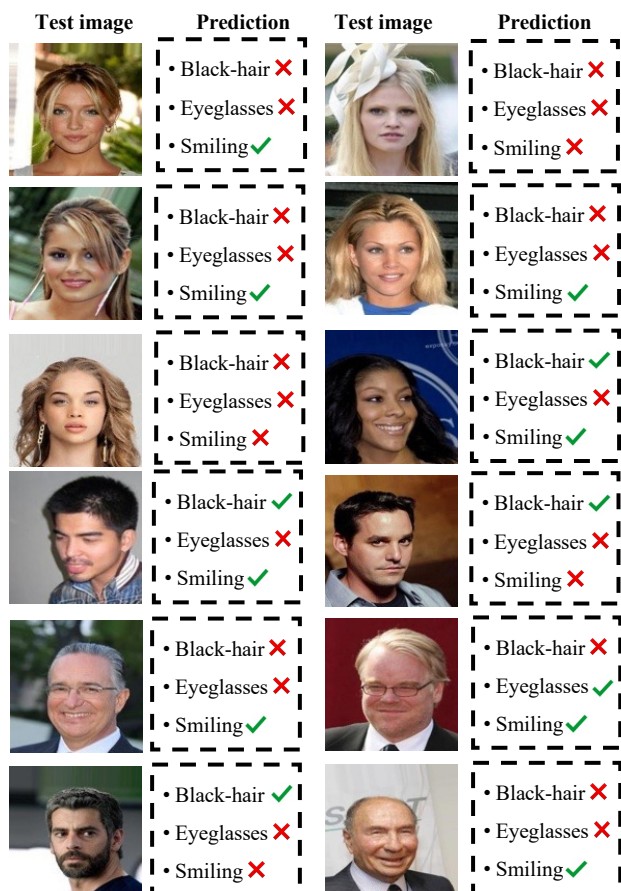

Figure X: **Facial Attribute Prediction Generalization.** Facial attribute prediction results on ClebaA images. Despite trained with only female faces from CelebA, our model can effectively generalize to new scenes containing male faces.

ure XIII. By employing multiple random initializations, our approach ensembles several starting points and corresponding optimization paths, achieving significant improvement over the capability to converge to the optimal solution. In Table IV, we further quantitatively demonstrate that as the number of random initialization starting points increases, the perception performance of our model improves.

**Prompt Weighting for Multi-Concept Perception.** We conducted an additional comparison to see whether naïve

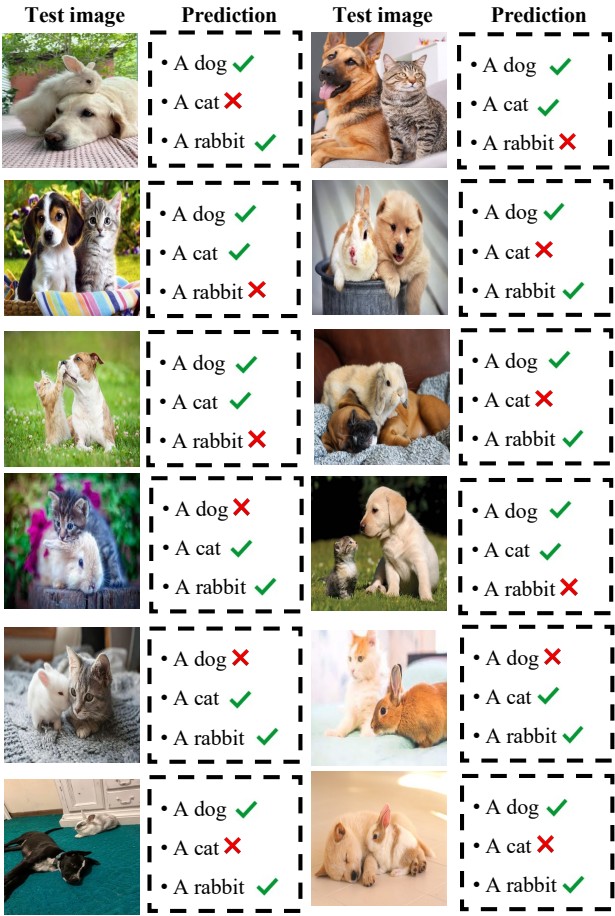

| Test image | Prediction | Test image | Prediction |
|---|---|---|---|

Figure XI: **Zero-Shot Multi-Object Perception.** Zero-shot multi-object perception results on natural images containing two animals from a finite set {dog, cat, rabbit}. By leveraging pretrained Stable Diffusion without any additional training, our model predicts object categories accurately.

prompt weighting could solve the zero-shot perception task, using the Compel package. Specifically, we applied prompt weighting to the compound prompts including "a photo of a cat++, a dog, and a rabbit", "a photo of a cat, a dog++, and a rabbit", and "a photo of a cat, a dog, and a rabbit++". The underlying idea is that if the image contains specific concepts (e.g., a cat and a dog), then the prompts "a photo of a cat++, a dog, and a rabbit" and "a photo of a cat, a dog++, and a rabbit" are expected to result in lower denoising error (higher likelihood) than the prompt "a photo of a cat, a dog, and a rabbit++". To determine which two objects present in the scene, we choose the two prompt weighted compound prompts that have the lowest denoising error. We report the zero-shot perception accuracy in Table V. The results suggest that simple prompt weighting may not be sufficient for effective multi-concept inference in this setting.

**Object Discovery on CLEVRTex.** Our model has been quantitatively evaluated on widely adopted datasets

---

**Algorithm 4** Concept Number Inference Algorithm

1: **Require:** an image $x$, trained denoising model $\epsilon_\theta$
2: Initialize denoising error list $\mathbf{E} = \text{zeros}(K_{max} - K_{min})$
3: **for** $K = K_{min}, \ldots K_{max}$ **do**
4:   ▷ *Initialize multiple (R) groups of concepts*
5:   Initialize concepts $\{c_r^1, c_r^2, \ldots, c_r^K\}_{r=1}^R \sim \mathcal{N}(0,1)$
6:   ▷ *Run Stochastic Gradient Descent*
7:   **for** $n = 1, \ldots, N_{\text{step}}$ **do**
8:     $\epsilon_n \sim \mathcal{N}(0,1), t_n \sim \text{Unif}(\{1, \ldots, T\})$
9:     $x^{t_n} = \sqrt{\bar{\alpha_{t_n}}} x + \sqrt{1 - \bar{\alpha_{t_n}}} \epsilon_n$
10:     $\Delta c_r^k \leftarrow \nabla_{c_r^k} \|\epsilon_n - \sum_{k=1}^K \epsilon_\theta(x^{t_n}, t_n, c_r^k)\|^2$
11:   **end for**
12:   ▷ *Evaluate denoising error for each configuration*
13:   Initialize a denoising error list $\mathbf{E_R} = \text{zeros}(R)$
14:   **for** $i = 1, \ldots, N_{\text{sample}}$ **do**
15:     $\epsilon_i \sim \mathcal{N}(0,1), t_i \sim \text{Unif}(\{1, \ldots, T\})$
16:     $x^{t_i} = \sqrt{\bar{\alpha_{t_i}}} x + \sqrt{1 - \bar{\alpha_{t_i}}} \epsilon_i$
17:     **for** $r = 1, \ldots, R$ **do**
18:       $\mathbf{E_R}[r] \mathrel{+}= \|\epsilon_i - \sum_{k=1}^K \epsilon_\theta(x^{t_i}, t_i, c_r^k)\|^2$
19:     **end for**
20:   **end for**
21:   ▷ *Select the group with lowest denoising error*
22:   $\mathbf{E}[K] = \min_{r \in \{1, \ldots, R\}} \frac{1}{N} \mathbf{E_R}[r]$
23: **end for**
  $\hat{K} = \text{argmin}_{K \in \{K_{min}, \ldots, K_{max}\}} \mathbf{E}[K]$
24: **return** $\hat{K}$

---

(CLEVR and CelebA) that are commonly used for object discovery and multi-label classification. While these datasets are relatively simple, they serve as strong benchmarks for measuring effectiveness and generalization. Nonetheless, we believe that testing our model on more complex scenes would be interesting. To take a first step towards that end, we conducted additional evaluations on ClevrTex (Karazija et al., 2021), which features diverse object colors, textures, shapes, and complex backgrounds. As shown in the Table VI, our model outperforms all baselines in object discovery on ClevrTex, demonstrating its potential scalability to more complex scenarios.

**Inference Time.** To evaluate inference efficiency, we conducted a comparison of runtime performance between our method and baseline models on an NVIDIA H100 GPU. We evaluate inference time for discrete concept inference on CelebA considering more attributes. As shown in Table VII, the inference time of our approach is comparable to the baseline model Generative Classifier. To further enable our model to work on a large number of concept settings (i.e., larger $K$), we developed a continuous approximation of our approach that allows gradient-based optimization, thereby avoiding the exponential ($M^K$) inference cost. The gradient-based discrete concept inference algorithm is outlined in

Figure XII: **Visualization of Object Number Inference in Object Discovery Tasks.** We train our model with CLEVR images containing 3-5 objects. During inference, given a test image, we try to use $K = 3, ..., 8$ object coordinates to fit the image through our inverse generative modeling approach following Algorithm 3 and Algorithm 4. We can see how the estimated coordinates mismatches ground truth coordinates when the object number differs from ground truth number.

| Models | OOD Accuracy ↑ |
|---|---|
| Diffusion Classifer | 70.4% |
| Compel | **35.2%** |
| Ours | 87.3% |

Table V: **Accuracy of Zero-Shot Perception with Prompt Weighting.** We evaluate the perception accuracy of the prompt weighting approach on the animal dataset and demonstrate that our approach significantly outperforms it.

Algorithm 5.

Specifically, to infer binary labels with gradient-based optimization, we relax the learnable binary labels to continuous parameters in the range (0, 1). These continuous parameters are optimized using gradient descent and clamped to (0, 1) at each iteration to remain valid. After optimization, we decide a label is 0 if the corresponding optimized relaxed parameter is smaller than 0.5, otherwise the label is 1. As is shown in Table VII, the continuous gradient-based approach reduces inference time significantly, making it scale linearly with the number of concepts ($O(K)$).

Additionally, we also provide the runtime of our approach and baselines on the zero-shot object perception task. Simi-lar to the previous setting, we have developed a continuous approximation for the zero-shot object perception task to improve inference efficiency. The gradient-based multi-object perception algorithm is outlined in Algorithm 6.

Specifically, for each concept (e.g., "a photo of a cat"), we assign a learnable weight to its corresponding noise prediction in the compositional model. These weights are then optimized via gradient descent. After optimization, we select the top two concepts with the highest optimized weights as the predicted objects in the scene. As shown in Table VIII this continuous relaxation leads to a significant reduction in inference time, with the time complexity scaling linearly with the number of candidate concepts ($O(K)$).

### A.3 Dataset Details

**Object Discovery.** We train our compositional generative model on 26132 CLEVR images (Johnson et al., 2017), each containing 3-5 objects of varying color, shape, size and texture. We resize the training images to a resolution of $64 \times 64$. For each image, the 2-D center coordinates of objects are also available, which are of continuous value and normalized between 0 and 1 by dividing the coordinates in terms of pixel value by the image resolution. The out-of-distribution test set consist of images with 6-8 objects either

| Models | In-distribution (CLEVRTex 3-5 objects) | | Out-of-distribution (CLEVRTex 6-8 objects) | |
|---|---|---|---|---|
| | Perception Rate ↑ | Estimation Error ↓ | Perception Rate ↑ | Estimation Error ↓ |
| ResNet-50 (He et al., 2016) | 3.9% | $2.0e^{-3}$ | 1.8% | $2.0e^{-3}$ |
| SlotAttn (Locatello et al., 2020) | 41.9% | $1.5e^{-3}$ | 35.2% | $1.6e^{-3}$ |
| GC (Li et al., 2024) | 69.6% | $9.8e^{-4}$ | 52.9% | $1.4e^{-3}$ |
| Ours | **85.2%** | $\mathbf{5.1}e^{-4}$ | **72.4%** | $\mathbf{7.8}e^{-4}$ |

Table VI: **Accuracy of Object Discovery.** Quantitative evaluation of object perception results on CLEVRTex for both in-distribution (3-5 objects) and out-of-distribution (6-8 objects) test settings. Perception rate and estimation error are reported. Our approach outperforms all the baselines, and the margin is especially significant for the out-of-distribution setting, demonstrating strong generalization capability.

| Models | OOD Accuracy ↑ | Inference Time ↓ |
|---|---|---|
| GC (Li et al., 2024) | 51% | 28.49s |
| IGM (Ours) | **60%** | 29.10s |
| IGM (Ours)- continuous approx | 55% | **22.15s** |

Table VII: **Accuracy and Inference Time of Global Factor Perception.** The inference time of both our approach and Generative Classifer are evaluated on CelebA considering 4 attributes including "black hair", "chubby", "eyeglasses", and "smiling". To avoid exponential computation cost for discrete concept inference through enumeration, we further develop a continuous approximation of our approach through gradient-based search algorithm, which significant reduces inference time while maintains generalization performance.

from the CLEVR dataset (Johnson et al., 2017), or from the CLEVRTex dataset (Karazija et al., 2021).

**Facial Feature Prediction.** For our facial feature experiments, we trained our model with 40612 *female face* images from the CelebA dataset (Liu et al., 2015). For each face image, three attributes are available including: Black Hair, Eyeglasses, and Smiling, each represented by categorical values $\{-1, 1\}$. We represent the attribute labels with one-hot encoding during training. The out-of-distribution test set consists of *male faces* only.

**Zero-Shot Multi-Object Perception.** For the zero-shot multi-object perception task, our approach directly leverages pretrained text-to-image generative models,requiring no additional training data. To evaluate the perception performance of our proposed approach in this task, we manually collected a small real-world dataset consisting of 71 random realistic images from the Internet, each containing two animals from $\{dog, cat, rabbit\}$. Specifically, this dataset consists of 20 images containing a cat and a dog, 22 images containing a cat and a rabbit, and 29 images containing a dog and a rabbit. In line with the common practice in text-to-image generative models, the prompt corresponding to these object concepts are: "a photo of cat", "a photo of dog", and "a photo of rabbit".

**Other Datasets.** Although our approach in this work focuses on the binary task of facial feature detection, it has the potential to handle more complex tasks, such as detecting edited facial attribute sequences explored in SeqDeepFake (Shao et al., 2022), which we leave for future work. Furthermore, in CelebA, the division between in-distribution and out-of-distribution data can also be defined such that

the in-distribution set includes all relevant factors but not all of their possible combinations, while the out-of-distribution set comprises novel combinations of these same factors (Schott et al., 2021; Wiedemer et al., 2023). However, inducing such combinatorial shifts in a controlled manner is not feasible with real-world datasets. Our primary goal is to evaluate the generalization ability of our compositional inference framework under challenging out-of-distribution (OOD) conditions by ensuring that the test data is substantially different from the training data. For example, in object discovery, the model is trained on Clevr while tested on a different dataset ClevrTex. Following the same spirit, we decided to train our model on CelebA female faces while testing it on CelebA male faces, as there are significant facial characteristic differences between the two groups.

### A.4 Training Details

We train a conditional latent diffusion model with latent space of 4 channels and resolution $8 \times 8$, which uses pretrained VAE to encode input images into the latent space. The latent space image is scaled with a factor of 0.18215. The denoising network adopts the Unet architecture (Ronneberger et al., 2015) as commonly used in diffusion models that takes the latent space image as input along with label conditioning and outputs noise predictions. Specifically, the input for the denoising network is of $8 \times 8$ and the cross attention dimension is 2 (the object coordinates dimension is 2) for object discovery and 6 (the one-hot encoding of facial attributes is of dimension 6) for facial feature prediction. We use 1000 diffusion steps and linear beta schedule for training. For other hyperparameters, we use a batch size 128 and a learning rate $2e^{-5}$.

| Models | OOD Accuracy ↑ | Inference Time ↓ |
|---|---|---|
| Diffusion Classifer (Li et al., 2023a) | 70% | 99.44s |
| IGM (Ours) | **87%** | 179.96s |
| IGM (Ours)- continuous approx | 75% | **101.05s** |

Table VIII: **Accuracy and Inference Time of Zero-Shot Multi-Object Perception.** The inference time of both our approach and Diffusion Classifer are evaluated on the zero-shot multi-object perception task. To avoid exponential computation cost for discrete concept inference through enumeration, we further develop a continuous approximation of our approach through gradient-based search algorithm, which significantly reduces inference time while maintains generalization performance.

**Test image** **Initialization 1** **Initialization 2** **Initialization 3**

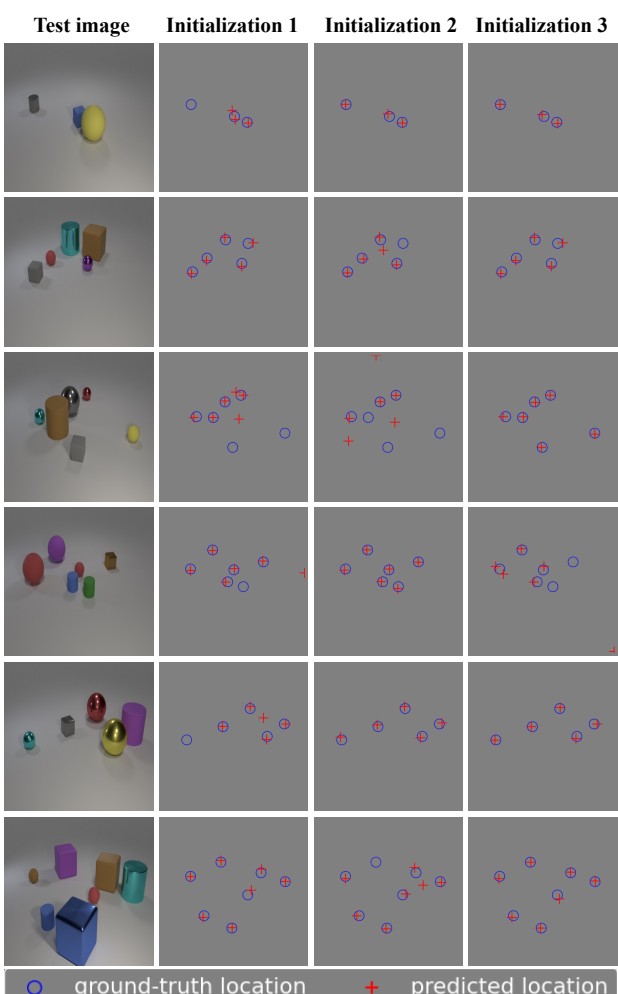

○ ground-truth location    + predicted location

Figure XIII: **Visualization of Random Concept Initialization in Object Discovery Tasks.** We train our model with CLEVR images containing 3-5 objects. During inference time, given a test image, we initialize object coordinates randomly and run stochastic gradient descent to iteratively refine the coordinates. Due to the non-convexity of the problem, the optimization procedure can converge either to the optimal solution or local minima, depending on initialization values. We propose to employ multiple random initializations, so that our approach can ensemble several starting points and corresponding optimization paths, achieving significant improvement over the capability to converge to the optimal solution.

---

**Algorithm 5** Gradient-based Discrete Concept Inference Algorithm

---

1: **Require:** an image $x$, trained denoising model $\epsilon_\theta$
2: Initialize relaxed continuous labels $l^1, ..., l^K \in (0, 1)$
3: ▷ *Construct psudo one-hot encoding for labels*
4: $c^1 = [l^1, 1 - l^1, 0, 0, 0..., 0, 0]$
5: $c^2 = [0, 0, l^2, 1 - l^2, 0..., 0, 0]$
6: ......
7: $c^K = [0, 0, 0, 0, 0..., l^K, 1 - l^K]$
8: ▷ *Run Stochastic Gradient Descent*
9: **for** $n = 1, \ldots, N_{\text{step}}$ **do**
10:     $\epsilon_n \sim \mathcal{N}(0,1), t_n \sim \text{Unif}(\{1, \ldots, T\})$
11:     $x^{t_n} = \sqrt{\bar{\alpha_{t_n}}} x + \sqrt{1 - \bar{\alpha_{t_n}}} \epsilon_n$
12:     $l^k \leftarrow l^k - \lambda \nabla_{l^k} \|\epsilon_n - \sum_{k=1}^{K} \epsilon_\theta(x^{t_n}, t_n, c^k)\|^2$
13:     ▷ *Clamp $l^k$ to (0, 1)*
14:     $l^k \leftarrow l^k.clamp(0, 1)$
15: **end for**
16: **if** $l^k < 0.5$ **then**
17:     $l^k \leftarrow 0$
18: **else**
19:     $l^k \leftarrow 1$
20: **end if**
21: **return** $l^1, l^2, ..., l^K$

---

### A.5 Baselines Details.

We compare our model against multiple discriminative and generative baselines. In this section, we introduce details on how these baselines are trained for scene understanding tasks considered in Section 4.

**ResNet-50 for Object Discovery.** For the object discovery tasks, the maximal number of objects in images is 5 in the training set and 8 in the test set. To enable ResNet-50 (He et al., 2016) to infer coordinates from images with 8 objects, we append a linear layer with input dimension 2048 and output dimension 16 on top of ResNet-50, followed by a sigmoid layer that outputs values between 0 and 1. The outputs is further reorganized into a $8 \times 2$ matrix with each row representing center coordinates of an object. We match the model output with ground truth object coordinates by minimizing the MSE loss to train the model. Since the dimension of ground truth coordinates in training data is $K \times 2$, where $3 < K < 5$, we pad additional $8 - K$ coordi-

---

**Algorithm 6** Gradient-based Zero-Shot Perception Algorithm

---

1: **Require:** an image $x$, trained denoising model $\epsilon_\theta$, $prompt^1 = $ "A photo of a cat", $prompt^2 = $ "A photo of a dog", $prompt^3 = $ "A photo of a rabbit"
2: ▷ *Get Text Embeddings for Prompts*
3: $c^k = \text{TextEmbedding}(prompt^k)$
4: Initialize concept weights $w^1, w^2, w^3$
5: ▷ *Run Stochastic Gradient Descent*
6: **for** $n = 1, \ldots, N_{\text{step}}$ **do**
7: $\quad \epsilon_n \sim \mathcal{N}(0,1), t_n \sim \text{Unif}(\{1, \ldots, T\})$
8: $\quad x^{t_n} = \sqrt{\overline{\alpha_{t_n}}}x + \sqrt{1 - \overline{\alpha_{t_n}}}\epsilon_n$
9: $\quad w^k \leftarrow w^k - \lambda\nabla\|\epsilon_n - \sum_{k=1}^K w^k\epsilon_\theta(x^{t_n}, t_n, c^k)\|^2$
10: **end for**
11: ▷ *Select the two indices with largest weights*
12: indices $= top2([w^1, w^2, w^3])$
13: **return** $c^{\text{indices}}$

---

nates with values $(1,1)$ representing empty coordinates (no object) to match the dimension of model output.

**Slot Attention for Object Discovery.**     Slot Attention (Locatello et al., 2020) is an unsupervised discriminative method for object discovery with strong generalization performance, which, however, only provides segmentation masks without giving the center coordinates of objects. For a fair comparison, we modify and train Slot Attention with object coordinates supervision. Specifically, Slot Attention learns a set of slots that compete with each other through cross attention mechanism to interpret a given image. These slots represent a high level description of objects in the image. To enable slot attention to predict object locations, instead of decoding slots into pixel components, we decode them into individual object coordinates. We supervise the decoded object coordinates outputs with ground truth coordinates by minimizing the MSE loss, where the coordinate matching is achieved with Hungarian Algorithm (Kuhn, 1955). To enable this supervised version of Slot Attention to be able to infer object coordinates from out-of-distribution images with object number reaching 8, we set the slot number to be 8. Since the number of ground truth coordinates in training data is $K$, where $3 < K < 5$, we pad additional $5 - K$ coordinates with values $(1,1)$ representing empty coordinates (no object) to match the dimension of model output.

**DINOSAUR for Object Discovery.**     DINOSAUR (Seitzer et al., 2022) is an enhanced extension of Slot Attention, which aggregates and reconstructs high-level semantic features extracted from the pretrained self-supervised DINO model, leading to improved object discovery performance. Like Slot Attention, DINOSAUR outputs only segmentation masks and does not provide object center coordinates. We adapt DINOSAUR for object discovery in the same manner

as described earlier for Slot Attention.

**Generative Classifier for Object Discovery.**     Generative Classifier (Li et al., 2024) is originally proposed to solve single-label classification problems by using diffusion models, where they try to find the categorical class that minimize denoising error. To infer object coordinates of continuous values, we adapt Generative Classifier by training a generative model that takes multiple object coordinates as conditioning. During inference, we can inverse the generative model and solve an optimization problems to find a set of object coordinates that best describe the image. For a fair comparison, we train this model following our proposed model. The only difference is that they train a single denoising network taking all coordinates as conditioning, while we train a set of denoising networks each taking an individual object coordinates as conditioning for compositional modeling. During inference, they can follow our proposed inference procedure in Algorithm 3.

**ResNet-50 for Facial Attribute Prediction.**     ResNet-50 (He et al., 2016) has been widely used to solve classification problems. It is straightforward to apply ResNet-50 to solve the facial attribute prediction task. We append a linear layer with input dimension $2048$ and output dimension $3$ on top of ResNet-50, followed by a sigmoid layer that probability values between 0 and 1. We supervise the model outputs with ground truth facial attribute labels by minimizing the BCE loss. During inference, ResNet-50 choose class label with high probability as classification results.

**Generative Classifier for Facial Attribute Prediction.** Generative Classifier (Li et al., 2024) originally can only solve single-label classification tasks. To enable Generative Classifier to perform multi-label classification, we train a diffusion model taking all three facial attributes as conditioning. During inference, we can enumerate through all possible facial attribute combinations (e.g., combination 1: "black hair, eyeglasses, smiling", combination 2: "not black hair, eyeglasses, smiling", etc.) and evaluate denoising errors. The one combination with smallest denoising error is selected as multi-label classification results. Again, how Generative Classifier in this case differs from our model lies in the lack of compositional modeling.

**Generative Classifier Variant for Facial Attribute Prediction.**     For Generative Classifier Variant, the training procedure is the same as Generative Classifier, but the inference procedure is different. In stead of enumerating attribute combinations, we can evaluate these attributes separately. Specifically, for attribute "black hair", we can evaluate the denoising error of "black hair" and "not black hair" conditioning and determine one of them with smaller denoising error as classification results. We then follow the same procedure to classify other attributes. This inference approach is desired to avoid unaffordable computation complexity

when the number of labels is very large.

**Diffusion Classifier for Multi-Object Perception.** Diffusion Classifier (Li et al., 2023a) is originally propose to solve zero-shot single-label classification problems by using pretrained text-to-image generative models without requiring any training. To adapt Diffusion Classifier for multi-label classification, we can feed Diffusion Classifier a prompt that describes a combination of multiple concepts as text conditioning and evaluate denoising error. For example, to determine if an image contains cat and dog in our task, Diffusion Classifier can evaluate the denoising error of the following prompts: "a photo of a cat and a dog", "a photo of a cat and a rabbit" and "a photo of a dog and a rabbit". The prompts with smallest denoising error is selected as the classification results.

**Diffusion Classifier Variant for Multi-Object Perception.** Diffusion Classifier Variant differs from Diffusion Classifier by evaluating each prompts separately. Given an image containing two animals from a finite set $\{\text{dog}, \text{cat}, \text{rabbit}\}$, Diffusion Classifier evaluates the denoising error of following prompts: "a photo of a dog", "a photo of a cat " and "a photo of a rabbit", and then choose the two prompts with smallest denoising error as multi-label classification results.

## A.6  Future Work

Extending our approach to dynamic or interactive scenes presents an exciting direction for future work. In the dynamic setting, the reconstruction objective can be reformulated to involve reconstructing an entire video of the target interaction, rather than a single static image. Each composed generative model can be conditioned on different aspects of the interaction, such as the identity of individual objects or agents within the environment. The inverse generative modeling procedure can then be used to consistently discover and track objects over time, even when they are temporarily occluded, as well as to infer the distinct behaviors of individual agents in the interactive scene.

Another interesting direction is to extend scene understanding, such as inferring a complete scene-graph. In this setting, each composed factor in our systems would correspond to an "edge" or relation between a pair of objects. We could then enumerate possible edges within the graph, seeking a combination whose composed set of relations yields an accurate reconstruction of the scene.

