# OpenReview forum: "Compositional Scene Understanding through Inverse Generative Modeling"
_ICML.cc/2025/Conference — ICML 2025 poster_

### Official Review · Reviewer_s7kA · 2025-03-14

**Overall Recommendation:** 4

**Summary:**

This work demonstrates how generative vision models can be composed to enable robust compositional scene understanding. The proposed solution enables different scene understanding tasks involving discrete or continuous factors, such as localization and multi-attribute classification. The method can also incorporate pretrained diffusion models to enable zero-shot classification.

**Claims And Evidence:**

1. The proposed method can accurately infer concepts.
    - The prediction of continuous factors is demonstrated well by the results in Tab. 1 and Figs. 4&5, where the method achieves impressive performance.
    - The prediction of discrete factors is demonstrated well in Tab. 5 and Fig. 6.
2. The proposed method can accurately infer the number of concepts.
    - This is supported well by Fig. 3.
3. The proposed method can incorporate pretrained diffusion models.
    - This is supported well by Fig. 7 and Tab. 3.

**Essential References Not Discussed:**

No.

**Experimental Designs Or Analyses:**

The experimental design for object localization using CLEVR and multi-attribute classification using CelebA is sound and uses established datasets.

- For the experiments on CelebA, I am wondering why the authors opt to use all male faces as the OOD set, which effectively removes the female/male attribute from classification. The more standard subdivision of the dataset in ID/OOD regions for compositional tasks would be to ensure that the ID set contains all values of all factors, but not all combinations, e.g., all women are only shown smiling, all men are only shown with a neutral face in the ID set. See, e.g., [1,2].
- What is the dataset used in Sec. 4.3 and Fig. 7/Tab. 3? Is this a dataset curated by the authors? A comparison on CelebA with pretrained models in zero-shot mode could have been equally insightful, or the authors could have opted for an established multi-label classification dataset.

[1] Schott et al., 2022, Visual Representation Learning Does Not Generalize Strongly Within the Same Domain

[2] Wiedemer et al., 2023, Compositional Generalization from First Principles

**Methods And Evaluation Criteria:**

- The proposed optimization scheme for determining $K$ and the set of $c_k$s is sound and explained well.
- The evaluation is sound and uses established standard metrics.

**Other Comments Or Suggestions:**

- small typo L198: ignore --> ignoring

**Other Strengths And Weaknesses:**

The paper is overall clear, well-written and easy to follow. The method is an interesting and novel extension of prior works, and experiments are straightforward and demonstrate well the efficacy of the method.

**Questions For Authors:**

- How feasible would it be to extend this method to more "involved" types of compositions, such as part-whole relationships or hierarchical compositions? Could this method (in principle, given unlimited compute) be extended to infer, e.g., a complete scene-graph of an input?

**Relation To Broader Scientific Literature:**

The basic compositional model follows prior work in the literature, as also noted by the authors. To the best of my knowledge, the inversion of this process is novel, as is the application to multi-target classification and localization.

**Theoretical Claims:**

There are no theoretical claims.

---

> ### Author Rebuttal · Authors · 2025-03-31
>
> We thank the reviewer for the positive feedback and comments! Please see our response below about your concerns.
>
> **1. For the experiments on CelebA, I am wondering why the authors opt to use all male faces as the OOD set, which effectively removes the female/male attribute from classification. The more standard subdivision of the dataset in ID/OOD regions for compositional tasks would be to ensure that the ID set contains all values of all factors, but not all combinations, e.g., all women are only shown smiling, all men are only shown with a neutral face in the ID set.  See, e.g., [1,2].**
>
> Thank you for your insightful question and the suggested references! Our primary goal is to evaluate the generalization ability of our compositional inference framework under challenging out-of-distribution (OOD) conditions by ensuring that the test data is substantially different from the training data. For example, in object discovery, the model is trained on Clevr while tested on a different dataset ClevrTex. Following the same spirit, we decided to train our model on CelebA female faces while testing it on CelebA male faces, as there are significant facial characteristic differences between the two groups. However, we agree that subdivision of the dataset in [1,2], where all attribute values are present in the training set but not all combinations, is also helpful for evaluating compositional generalization, and we will update the paper to discuss this evaluation procedure. Unfortunately, this type of combinational shift is not available to gather in a controlled manner in real-world datasets, leading us to use the CelebA dataset.
>
> [1] Schott et al., 2022, Visual Representation Learning Does Not Generalize Strongly Within the Same Domain
>
> [2] Wiedemer et al., 2023, Compositional Generalization from First Principles
>
>
> **2. What is the dataset used in Sec. 4.3 and Fig. 7/Tab. 3? Is this a dataset curated by the authors? A comparison on CelebA with pretrained models in zero-shot mode could have been equally insightful, or the authors could have opted for an established multi-label classification dataset.**
>
> Thank you for the insightful question! Yes, the dataset used in Sec. 4.3, Fig. 7, and Table 3 was curated by the authors and consists of 20 images containing a cat and a dog, 22 images containing a cat and a rabbit, and 28 images containing a dog and a rabbit. Our primary focus in these experiments was to evaluate multi-object perception performance (local factor discovery) using compositional pretrained models compared against Diffusion Classifier. We agree that testing these models on CelebA to discover global attribute factors could provide additional insights and will do so in the final version of the paper.
>
> **3. Small typo L198: ignore --> ignoring**
>
> Thank you for pointing out the typo! We will fix it in the next version.
>
>
> **4. How feasible would it be to extend this method to more "involved" types of compositions, such as part-whole relationships or hierarchical compositions? Could this method (in principle, given unlimited compute) be extended to infer, e.g., a complete scene-graph of an input?**
>
> Thank you for the insightful question! We believe that our method would in principle be able to scale to more “involved” types of compositions, such as inferring a complete scene-graph. In the setting of a scene graph, each of the composed factors in our systems would correspond to an “edge” or relation between two objects. We then enumerate through different edges in a graph until we get a graph whose composed set of edges leads to a good reconstruction of the scene. We will update the paper and discuss this in the future work.

---

### Official Review · Reviewer_Tv9z · 2025-03-14

**Overall Recommendation:** 4

**Summary:**

The paper introduces an innovative inverse generative modeling framework designed to perform compositional scene understanding. Its primary innovation is to interpret visual scenes as compositions of smaller generative models, enabling effective generalization to complex or unseen scenes. Specifically, the authors propose training a generative model through diffusion processes and then using this model inversely, inferring conditional parameters (eg, object categories, locations, facial attributes) from images. Major contributions include the introduction of compositional modeling into generative inference, achieving improvement in generalization capability, and a demonstration of the method's broad applicability across various scene understanding tasks, including object discovery and zero-shot perception.

**Claims And Evidence:**

The claims made in the paper, primarily that compositional inverse generative modeling improves generalization on various visual tasks, are well supported by clear experimental evidence.
The experiments across diverse tasks (ie, object detection on CLEVR and CLEVRTex datasets, facial attribute classification on CelebA, and zero-shot object recognition using pretrained Stable Diffusion) demonstrate improved performance over multiple baselines.

**Essential References Not Discussed:**

n/a

**Experimental Designs Or Analyses:**

The experimental design and analyses presented in the paper are sound and robust.
The authors differentiate between in-distribution and out-of-distribution scenarios, clearly showcasing generalization.
The ablation studies, such as testing the effects of the multiple-initialization strategy, further strengthen the validity and thoroughness of their experiments.

**Methods And Evaluation Criteria:**

The proposed method and evaluation criteria are appropriate for the addressed problems.
Using established benchmarks (CLEVR, CLEVRTex, CelebA) and pretrained diffusion models (Stable Diffusion) provides relevant metrics, such as perception rate, estimation errors, and classification accuracy, all of which clearly measure method effectiveness and generalization.

**Other Comments Or Suggestions:**

Minor suggestions include:
1. Clarify computational overhead or potential methods to reduce complexity in real-world deployments.
2. Consider discussing the limitations of compositional approximations explicitly in the paper.

**Other Strengths And Weaknesses:**

Strengths:
1. The novelty of applying compositionality explicitly within inverse generative modeling.
2. Significant and demonstrable improvements over strong baseline models, especially in generalization scenarios.
3. Effective demonstration of zero-shot capabilities using pretrained generative models, indicating high practical relevance.

Weaknesses:
1. The computational complexity involved when enumerating concept combinations might limit scalability for tasks involving numerous discrete concepts.

**Questions For Authors:**

1. How scalable is the compositional inverse generative modeling approach when dealing with a very large number of visual concepts or categories? Would an alternative optimization strategy reduce complexity without sacrificing accuracy?
2. Could you further elaborate on how this compositional modeling approach would handle dynamic or interactive scenes (eg, videos)? Have you considered any temporal extensions or adaptations?

**Relation To Broader Scientific Literature:**

This paper effectively builds on prior work in generative modeling (diffusion models, generative classifiers) and compositional visual understanding.

**Theoretical Claims:**

No explicit theoretical proofs were presented.

---

> ### Author Rebuttal · Authors · 2025-03-31
>
> We thank the reviewer for the positive feedback and comments! Please see our response below about your concerns.
>
> **1.How scalable is the compositional inverse generative modeling approach when dealing with a very large number of visual concepts or categories? Would an alternative optimization strategy reduce complexity without sacrificing accuracy? Clarify computational overhead or potential methods to reduce complexity in real-world deployments.**
>
> Thank you for the insightful question! Overall, we agree that our approach is substantially slower than existing feedforward methods, but we believe that the additional computational cost is justified by the ability of our to generalize well to unseen complex scenes. When dealing with a large number of visual concepts and categories, our naive algorithm of enumeration has exponential growth of the search space ($M^K$) for discrete values. Beyond the early stopping strategy discussed in our submission, several additional approaches can potentially significantly mitigate this computational bottleneck. One approach could be to use heuristic search algorithms on discrete values – for instance we can run beam search with a beam width of K over each attribute sequentially  which can reduce time complexity from $M^K$ to $O(M*K)$, making inference more efficient for large discrete spaces. Alternatively, continuous relaxation of discrete variables with gumbel-softmax/concrete relaxation could allow the use of gradient-based search and thus avoid enumerating all configurations. Finally, since our approach allows parallel processing across the $M^K$ configurations, inference time can be drastically reduced given sufficient computational resources, potentially approaching the time required for a single configuration evaluation. We will discuss these possible extensions in the future work.
>
>
> **2. Consider discussing the limitations of compositional approximations explicitly in the paper.**
>
> Thank you for the valuable suggestion! We will add a limitations section in the next version of our paper and discuss this. Our compositional generative modeling approach assumes object concept independence given the input image, enabling combinatorial generalization beyond the training distribution. However, one possible limitation of this full independence approximation is that it ignores the interaction between objects, which are crucial in many real-world scenarios. As a remedy, we could potentially learn additional models that model interactions between object components which can also be composed to represent more complex scenes.  We will explicitly discuss this limitation and potential remedies in the limitations section.
>
> **3. Could you further elaborate on how this compositional modeling approach would handle dynamic or interactive scenes (eg, videos)? Have you considered any temporal extensions or adaptations?**
>
> Thank you for the insightful question! Extending our approach to dynamic or interactive scenes is an exciting direction for future work that we are also currently exploring. In the dynamic setting, we can change the reconstruction objective to not be reconstructing a given image but instead to reconstruct the entire video of the target interaction. We can condition each composed generative model on different aspects of the interactions such as the identity of an individual objects or agent in the environment. The inverse generative modeling procedure can then be used to persistently discover objects in the scene, even if they are occluded at different points in time, or to infer the individual behaviors of each individual agent in the interactive environment. We will update the paper and discuss this in future work.

---

### Official Review · Reviewer_xCEx · 2025-03-17

**Overall Recommendation:** 3

**Summary:**

This paper casts scene understanding as an inverse generative modeling problem, where attributes of the scene are extracted by seeking parameters of the generative model that best fit a given image. To facilitate generalizability, the paper proposes to model visual scenes compositionally - by model sub components of the scene with smaller models, and compose them in the form of energy based functions. The paper proposes different optimization method to search for discrete and continuous attribute variables, and experimental results show that the proposed formulation can more accurately find attributes of images comparing to related methods, while being able to generalize to novel scenes that have different content distribution than training images.

**Claims And Evidence:**

The claim that compositional inverse generative modelling enables more effective scene understanding is supported empirically by local factor perception (object counting, position estimation on CLEVR, CLEVREX datasets), global factor perception (facial attributes prediction on CelebA dataset), and zero-shot multi-object perception, of which the proposed method show better results than related previous methods.
But the datasets used in the experiments mainly contain simple concepts (CLEVR dataset with limited number of objects, or human face of a fixed set of attributes), as also mentioned in the limitation section of the paper, it's no practical to scale the proposed method (in its current state) to more complex, realistic real world scenes.

**Essential References Not Discussed:**

I haven't found missing essential references.

**Experimental Designs Or Analyses:**

Yes, I checked all the subsections of the experiment section. No obvious issues.

**Methods And Evaluation Criteria:**

method:
1. The compositional generative modeling formulation of the proposed method makes intuitive sense. However, in order to escape local optima, the success inference of continuous concepts relies on multiple random trials. The number of random trials may vary depending on the complexity of the concept, which may not guarantee that the method will succeed in problems that are significantly different than the tested problems in the paper without further tuning this hyperparameter.

datasets:
1. The tested datasets contain simple concepts (limited number of objects in CLEVR, limited set of attributes in face), so although performance evaluation positively supports the proposed idea, the method in its current state still cannot be extended to more complex real world scenes.

metrics:
1. As mentioned in the limitations in the paper, the searching process of the proposed method is long due to exhaustive search nature, so when comparing with related methods, in addition to the perception accuracy, it'd be better to add inference time comparison as well.

**Other Comments Or Suggestions:**

please see questions

**Other Strengths And Weaknesses:**

Strengths:
1. The paper is well written and easy the follow
2. The description of experimental setups are is comprehensive

**Questions For Authors:**

1. When inferencing a continuous concept, how to decide how many random trials are needed in order to escape a local optima solution?

2. For the tested perception tasks, how is the inference time of the proposed method compared to the related baselines ?

**Relation To Broader Scientific Literature:**

The proposed method approximates the conditional probability distribution of the scene under different concepts with product of probability under individual concept, this is related to [1]; It further uses the denoising function in diffusion model to estimate the gradient of energy function, which is inspired by [2]






[1] Du, Yilun, and Leslie Kaelbling. "Compositional generative modeling: A single model is not all you need." arXiv preprint arXiv:2402.01103 (2024).
[2] Liu, Nan, et al. "Compositional visual generation with composable diffusion models." European Conference on Computer Vision. Cham: Springer Nature Switzerland, 2022.

**Theoretical Claims:**

There are no theoretical claims in this paper.

---

> ### Author Rebuttal · Authors · 2025-03-31
>
> We thank the reviewer for the constructive feedback and comments! Please see our response below about your concerns.
>
> **1. The tested datasets contain simple concepts (limited number of objects in CLEVR, limited set of attributes in face), so although performance evaluation positively supports the proposed idea, the method in its current state still cannot be extended to more complex real world scenes.**
>
> Thank you for your insightful question! Our model is evaluated on widely adopted datasets (CLEVR and CelebA) that are commonly used for object discovery and multi-label classification. While these datasets are relatively simple, they serve as strong benchmarks for measuring effectiveness and generalization, as also acknowledged by reviewers 6fxB, Tv9z and s7kA. Nonetheless, we agree that testing our model on more complex scenes would be interesting. To take a first step towards that end, we conducted additional evaluations on ClevrTex [1], which features diverse object colors, textures, shapes, and complex backgrounds. As shown in the table below, our model outperforms all baselines in object discovery on ClevrTex, demonstrating its potential scalability to more complex scenarios.
>
> |                  | In-distribution |(ClevrTex 3-5 objects)| Out-of-distribution |( ClevrTex 6-8 objects)|
> |:---:|:---:|:---:|:---:|:---:|
> |                        | Perception Rate | Estimation Error | Perception Rate | Estimation Error |
> | **ResNet-50**          | 3.9             | 0.00196          | 1.8             | 0.00198          |
> | **Slot Attention**     | 41.9            | 0.00152          | 35.2            | 0.00161          |
> | **Generative Classifier** | 69.6         | 0.00098          | 52.9            | 0.00135          |
> | **Ours**              | 85.2            | 0.00051          | 72.4            | 0.00078          |
>
> [1] Karazija, et al., 2021 Clevrtex: A texture-rich benchmark for unsupervised multi-object segmentation.
>
> **2. When inferencing a continuous concept, how to decide how many random trials are needed in order to escape a local optima solution?**
>
> Thank you for raising this important point! We acknowledge that escaping local optima in continuous concept inference can be influenced by the number of random trials, as demonstrated by the ablation study (Table IV) of our submission. To determine the optimal number of trials, a principled approach is to monitor the reconstruction  error of the input image: if increasing the number of trials no longer leads to significant reconstruction improvements, further trials become unnecessary. In practice, our experiments show that a moderate number of trials (5 or 10) is sufficient for continuous inference tasks.
>
> **3. For the tested perception tasks, how is the inference time of the proposed method compared to the related baselines ?**
>
> Thank you for suggesting the evaluation metric! We run our model and baselines on Nvidia H100 and report inference times in the table below, where “N/A” indicates that the approach is not applicable to the task. Our results show that generative approaches, including our approach and Diffusion Classifier, require more time than the discriminative approaches like ResNet-50 and Slot Attention, while our approach consumes more time than Diffusion Classifier. Specifically, our approach takes longer than Diffusion Classifier due to the need to evaluate a composition of multiple diffusion models. However, this increased inference time for our approach comes with significantly improved generalization performance. Additionally, inference time bottleneck could be further mitigated through parallel processing, which our algorithm supports, or using heuristic search algorithms such as beam search to speed up optimization, which we leave for future work. We appreciate your suggestion and will include the inference time metric in the next version of our paper and discuss how improving inference speed is an interesting direction of future work.
>
> |                           | Local Factor (Clevr) | Global Factor (CelebA) | Zero-Shot (Pretrained)|
> |:---:|:---:|:---:|:---:|
> | ResNet-50     | 0.011s  | 0.023s |   N/A  |
> | Slot Attention     | 0.009s    |   N/A   |   N/A  |
> | Generative Classifier     |   102.874s    |    28.485s   |   99.436s   |
> | Ours     |    146.854s   |    29.095s    |   179.957s  |

---

### Official Review · Reviewer_6fxB · 2025-03-17

**Overall Recommendation:** 2

**Summary:**

This paper proposed an inverse generative modeling approach to understand attributes of a single image. This approach trains conditional generative models based on different conditions to construct a multi-conditional generative model through the composition of EBMs. The compositional modeling can generalize the generation ability to out-of-distribution images to generate combinations of conditions that have not been seen during the training phase. After the compositional generative model is trained, the inverse generative modeling algorithm is used to search for condition parameters that maximize the log-likelihood from a single input image as the inference results. For conditions with continuous attribute values, this paper proposed a gradient-based heuristic algorithm that can effectively search for optimal conditions in continuous space. The experiments show that the proposed method outperforms the comparative methods in tasks such as scene object number inference and object localization.

**Claims And Evidence:**

The claims made in this paper are clear, which are supported by the experiments.

**Essential References Not Discussed:**

The introduction to related works is generally complete, but I am not sure if there are any important recent works that have not been discussed.

**Experimental Designs Or Analyses:**

The experimental design and analysis are reasonable and effective.

**Methods And Evaluation Criteria:**

The datasets and evaluation criteria of this paper make sense for the problem.

**Other Comments Or Suggestions:**

In Line 8 of Algorithm 3, after calculating the gradient, should we need to update the original c_r^k?

In the reference, some formats are different from other places, e.g., the paper link in Lines 440-441.

**Other Strengths And Weaknesses:**

Strengths

The inverse generative modeling algorithm proposed in this paper does not require additional model training for image attribute inference. This makes the method have potential for direct application on existing compositional generative models. The experiments on multiple image understanding tasks also show the effectiveness and versatility of the approach.

Weaknesses

The compositional generative models are trained based on given conditions and images, so the proposed approach requires annotations of the corresponding image attributes. But Slot Attention is trained by reconstruction loss, which does not require the annotations. Therefore, one concern is whether the fairness of model comparison. In addition, I recommend the authors to compare with more advanced object-centric representation methods. For example, DINOSAUR [1], which is an extension of Slot Attention that uses the representation from more powerful pre-trained visual models.

One concern is about the efficiency of the inverse generative modeling algorithm. If the number of conditions or the number of selectable discrete values of each condition are large, the size of the search space M^K will increase exponentially. It will be valuable to conduct comparative experiments on the inference efficiency of models.

[1] Seitzer, Maximilian, et al. "Bridging the gap to real-world object-centric learning.", ICLR 2023

**Questions For Authors:**

In Table 3, why does the Diffusion Classifer Variant perform worse than the original Diffusion Classifer?

Could the authors explain how to implement object discovery based on Slot Attention? Although Appendix A.5 mentions that the supervised version of Slot Attention is used in the experiments, I wonder if it is possible to infer the number of objects and locate the objects only through the mask of each object extracted by Slot Attention.

**Relation To Broader Scientific Literature:**

The main contribution of this paper is to design a novel inverse generative modeling algorithm, which infers attributes of a single image by searching optimal conditional parameters on a trained compositional generative model. The training of compositional generative models is based on several previous approaches.

**Theoretical Claims:**

I checked the correctness of Equations 1-9. There is no proof in this paper.

---

> ### Author Rebuttal · Authors · 2025-03-31
>
> We thank the reviewer for the constructive feedback and comments! Please see our response below about your concerns.
>
> **1. One concern is the fairness of comparison with unsupervised Slot Attention. I recommend to compare with more advanced DINOSAUR.**
>
> Thank you for the insightful suggestion! We would like to clarify that the Slot Attention baseline we compare to in the paper has incorporated supervision on each head to the object coordinates in the scene (discussed in Appendix A.5), and thus has the same information as our approach. Furthermore, following the reviewer’s suggestion, we compared our model with DINOSAUR also with supervision on Clevr. As shown in the table below, DINOSAUR works better than Slot Attention, while our model outperforms DINOSAUR.
>
> ||In-distribution|(3-5 objects)|Out-of-distribution|(6-8 objects)|
> |:---:|:---:|:---:|:---:|:---:|
> ||Perception Rate|Estimation Error|Perception Rate|Estimation Error|
> |Slot Attention|80.4|0.00087|53.3|0.00130|
> |DINOSAUR|82.5|0.00084|59.0|0.00120|
> |Ours|94.7|0.00014|85.3|0.00035|
>
> **2. One concern is the efficiency of the algorithm. For discrete values, the size of the search space $M^K$ will increase exponentially.**
>
> Thank you for the insightful question! Overall, we agree that our approach is substantially slower than existing feedforward methods, but we believe that the additional computational cost is justified by the ability of our method to generalize well to unseen complex scenes. It’s true that our naive algorithm has exponential growth of the search space ($M^K$) for discrete values. Beyond the early stopping strategy discussed in our paper, several additional approaches can potentially significantly mitigate this computational bottleneck. One approach could be to use heuristic search algorithms on discrete values – for instance we can run beam search with a beam width of K over each attribute sequentially, which can reduce time complexity to $O(M*K)$, making inference more efficient for large discrete spaces. Alternatively, continuous relaxation of discrete variables with gumbel-softmax/concrete relaxation could allow the use of gradient-based search and thus avoid enumerating all configurations. Finally, since our approach allows parallel processing across the $M^K$ configurations, inference time can be drastically reduced given sufficient computational resources, potentially approaching the time required for a single configuration evaluation. We appreciate your suggestion, and will include comparative experiments of the overall time complexity in our paper and discuss ways to speed up inference.
>
> **3. In Algorithm 3, should we need to update the original $c_r^k$?**
>
> Yes, we should update $c_r^k$ with the gradient $\Delta c_r^k$ in line 8. We will add an explicit update step ($c_r^k ← c_r^k - \lambda\Delta c_r^k)$  to Algorithm 3 in the next version.
>
> **4. In the reference, some formats are different from others, e.g., Lines 440-441.**
>
> Thank you for pointing out the reference format issue! We will ensure reference format consistency in the next version.
>
> **5. In Table 3, why Diffusion Classifier Variant performs worse than the original Diffusion Classifier?**
>
> Diffusion Classifier is originally designed for single-label classification tasks and thus  performs more poorly  on the multi-object tasks we consider.  To apply Diffusion Classifier on our setting, we directly condition the generative model using compound prompts (e.g., “a photo of a dog and a cat”), which we refer to as the Diffusion Classifier baseline in our experiments. We also further explore a single object variant of Diffusion Classifier for this compound setting, where we condition the generative model on individual concepts (e.g., “a photo of a dog”) and select the two classes with the highest score. Details of both methods are detailed in A.5. Since the Diffusion Classifier Variant uses individual concepts to fit multi-concept images, it has worse performance than Diffusion Classifier baseline.
>
> **6. How to implement object discovery with Slot Attention? Is it possible to infer object number and locate objects only through masks extracted by Slot Attention.**
>
> Thank you for the insightful question! To enable slot attention to predict object locations with supervision, we decode slot representations into object coordinates and enforce a coordinate prediction MSE loss by comparing the decoded object coordinates with ground truth, where we use Hungarian Algorithm to align decoded coordinates and ground truth. To infer object number, during training, we can enforce empty slots to predict a fixed coordinate [1,1] (the normalized rightmost corner of the image), while object slots predict the actual object coordinates. At inference time, if the inferred coordinates are greater than a threshold close to [1,1], we determine these slots as empty, while the rest are considered to contain objects. This approach allows Slot Attention to infer both object number and their locations.

---

> > ### Comment · Reviewer_6fxB · 2025-04-05
> >
> > Thank you for the detailed response. The authors claim that while the proposed approach is slower, it offers stronger generalization to unseen scenes, which appears to be a trade-off. But I think this statement holds only if the computational cost remains within a reasonable range (not very higher than other methods). So far I do not see experimental results of computational efficiency that can substantiate the claim. The application of the approach will be limited if it exhibits exponential growth in complexity. So I would like to maintain my score.

---

> > > ### Author Response · Authors · 2025-04-06
> > >
> > > Thank you for engaging in this discussion period! We truly appreciate your time and constructive comments! To evaluate inference efficiency, we conducted a comparison of runtime performance between our method and baseline models on an NVIDIA H100 GPU. The table below shows the inference time for the discrete concept inference on CelebA considering 4 attributes. As shown, the inference time of our approach is comparable to the baseline model Diffusion Classifier. To further enable our model to work on a large number of concept settings (i.e., larger $K$), we developed a continuous approximation of our approach that allows gradient-based optimization, thereby avoiding the exponential ($M^K$) inference cost. We provide a link for the algorithm here (https://imgur.com/a/gpULhuR).
> > >
> > > Specifically, to infer binary labels with gradient-based optimization, we relax the learnable binary labels to continuous parameters in the range (0, 1). These continuous parameters are optimized using gradient descent and clamped to (0, 1) at each step to remain valid. After optimization, we decide a label is 0 if the corresponding optimized relaxed parameter is smaller than 0.5, otherwise the label is 1. As is shown in the table below, the continuous gradient-based approach reduces inference time significantly, making it scale linearly with the number of concepts ($O(K)$).
> > >
> > > | Models     | OOD Accuracy (CelebA 4 features) | Inference time (CelebA 4 features) |
> > > | ----------- | ----------- |  ----------- |
> > > | Diffusion Classifier     | 0.51      | 28.49s |
> > > | Ours  | 0.60       |   29.10s|
> > > | Ours (continuous approx)  | 0.55       |  22.15s |
> > >
> > > Additionally, we also provide the runtime of our approach and baselines on the zero-shot object perception task. Similar to the previous setting, we have developed a continuous approximation for the zero-shot object perception task to improve inference efficiency. The algorithm can be found in (https://imgur.com/a/SzAn0Jx).
> > >
> > > Specifically, for each concept (e.g., “a photo of a cat”), we assign a learnable weight  to its corresponding noise prediction in the compositional model. These weights are then optimized via gradient descent. After optimization, we select the top two concepts with the highest optimized weights as the predicted objects in the scene.  As shown in the table below, this continuous relaxation leads to a significant reduction in inference time, with the time complexity scaling linearly with the number of candidate concepts ($O(K)$).
> > >
> > > | Models     | Accuracy (Zero-Shot)| Inference Time  (Zero-Shot) |
> > > | ----------- | ----------- |  ----------- |
> > > | Diffusion Classifier     | 0.64      | 99.44s |
> > > | Ours  | 0.80       |   179.96s|
> > > | Ours (continuous approx)  | 0.68      |  101.05s|

---

### Official Review · Reviewer_7Q2x · 2025-03-17

**Overall Recommendation:** 4

**Summary:**

This paper presents a computational framework for mining the structural properties of natural scene images by recasting the problem as an inverse generative modeling task. Specifically, the authors propose a generic inverse generative modeling paradigm that integrates compositionality into a diffusion-based generative model, enabling robust generalization beyond the training distribution. During training, the model undergoes optimization by minimizing the average denoising error across discrete concepts. At inference, the framework performs a constrained search within a predefined range of concept cardinalities to identify the configuration that minimizes the average denoising error. For continuous concepts, the authors employ multiple randomized initialization points, iteratively discarding low log-likelihood candidates until convergence to a single optimal solution. Furthermore, they leverage stochastic gradient descent (SGD) to iteratively refine concept representations, enhancing computational efficiency. Empirical evaluations substantiate the efficacy of the proposed methodology, demonstrating substantial performance improvements across three scene understanding tasks—local factor perception, global factor perception, and zero-shot multi-object perception—on the CLEVR, CelebA, and a custom small-animal dataset. These results validate the proposed approach’s effectiveness and establish new state-of-the-art performance benchmarks in the domain of compositional scene understanding.

**Claims And Evidence:**

- In general, the claims of this paper are supported by clear and convincing evidence; however, two of the experiments are too trivial to fully validate the proposed method's claims.

**Essential References Not Discussed:**

- In general, this paper adequately references related approaches.

**Experimental Designs Or Analyses:**

- As mentioned above, the experiments on CelebA and the custom small-animal dataset are too trivial. For CelebA, it may be more appropriate to use the dataset from [1], which contains more attributes and is generated by StyleGAN. For the custom small-animal dataset, providing statistical characteristics such as distribution and ensuring a more consistent and diverse set of categories would make the evaluation more convincing.

[1] SeqDeepFake: Detecting and Recovering Sequential DeepFake Manipulation

**Methods And Evaluation Criteria:**

- Evaluating the proposed methods on CelebA and the custom small-animal dataset may not be convincing, as the images contain only a limited number of attributes (typically 2–3).

**Other Comments Or Suggestions:**

- In line 146, it would be clearer to use the full name, 'Energy-Based Model,' instead of the abbreviation when mentioning it for the first time.

**Other Strengths And Weaknesses:**

- The implementation of the approach is methodical and straightforward, enhancing its practical applicability.
- Comprehensive implementation details significantly improve the reproducibility of the research.
- However, certain aspects lack sufficient discussion, such as the relationship between image captioning models like BLIP-2, which undermines the novelty and contribution of the work.

**Questions For Authors:**

- 1. How does the proposed method perform on tasks involving a greater number of attributes, as mentioned above?
- 2. The authors should clarify the advantages of their approach compared to existing image captioning models.
- 3. In the zero-shot tasks, why does the diffusion classifier variant perform worse than the original version? Can the authors explain this phenomenon and provide a theoretical rationale for its relationship with the proposed method?

**Relation To Broader Scientific Literature:**

- This method provides a framework for mining the structural properties of natural scene images. Unlike existing approaches that rely on text-to-image models, the proposed method leverages generative models with flexible conditioning, allowing it to be applied to a broader range of visual understanding tasks. This opens up a new direction for scene understanding.

**Theoretical Claims:**

- This paper does not present many theoretical claims. Furthermore, the proposed method is simply an extension of an existing approach

---

> ### Author Rebuttal · Authors · 2025-03-31
>
> We thank the reviewer for the constructive feedback and comments! Please see our response below about your concerns.
>
> **1. CelebA images contain only a limited number of attributes (typically 2–3). How does the proposed method perform on tasks involving a greater number of attributes?**
>
> Thank you for the insightful question! To demonstrate the effectiveness of our method on a greater number of attributes, we conducted an experiment on CelebA using 4 attributes (Black Hair, Eyeglasses, Smiling, and Wearing Hat), given our time and computational constraints. As shown in the table below, our model consistently outperforms baseline models in both in-distribution and out-of-distribution accuracy. These results demonstrate the potential of our approach to handle more than just 2–3 attributes effectively.
>
> |                           | In-distribution Accuracy (Female only) | Out-of-distribution Accuracy (Male only) |
> |:---:|:---:|:---:|
> | ResNet-50 Classifier      | 0.76                                   | 0.57                                    |
> | Generative Classifier     | 0.78                                   | 0.51                                    |
> | Ours                      | 0.78                                   | 0.60                                    |
>
>
> **2. For CelebA, it may be more appropriate to use the dataset from SeqDeepFake.**
>
> Thank you for suggesting this interesting paper and dataset! After reviewing SeqDeepFake, we don’t think   that this dataset is directly applicable for our binary multi-label classification task. Our method requires binary labels for a fixed set of attributes, whereas SeqDeepFake provides edited attribute sequences with varying attribute annotation across images. Nevertheless, we find that the image captioning task in SeqDeepFake is very related and interesting, and would like to explore if our model can detect edited attribute sequences in future work. We will discuss the relevance of SeqDeepFake in the related work section in the next version of our paper.
>
> **3. For the custom small-animal dataset, providing statistical characteristics such as distribution and ensuring a more consistent and diverse set of categories would make the evaluation more convincing.**
>
> Thank you for your helpful suggestion! To clarify, our custom small-animal dataset consists of 20 images containing a cat and a dog, 22 images containing a cat and a rabbit, and 28 images containing a dog and a rabbit. We will include this dataset distribution description in the next version to enhance transparency and improve the evaluation's clarity.
>
> **4. Lack sufficient discussion on the relationship between image captioning models like BLIP-2. The authors should clarify the advantages of their approach compared to existing image captioning models.**
>
> Thank you for highlighting this connection to related work! By using pretrained text-to-image generative models (e.g., Stable Diffusion), our model can tackle image captioning tasks like BLIP-2. However, our approach is applicable to a broader range of scene understanding tasks other than image captioning. For example, by conditioning on object coordinates, our approach can perform object discovery tasks and even enable generalization to more complex scenes (many more objects) than seen at training. This flexibility and generalizability distinguishes our approach from traditional image captioning models. We appreciate your suggestion and will include a more detailed discussion in the next version.
>
>  **5. In line 146, it would be clearer to use the full name, 'Energy-Based Model,' instead of the abbreviation when mentioning it for the first time.**
>
> Thank you for pointing out the abbreviation issue! We will fix it in the next version.
>
> **6. In the zero-shot tasks, why does the diffusion classifier variant perform worse than the original version?**
>
> Diffusion Classifier is originally designed for single-label classification tasks and thus  performs more poorly  on the multi-object tasks we consider.  To apply Diffusion Classifier on our setting, we directly condition the generative model using compound prompts (e.g., “a photo of a dog and a cat”), as detailed in A.5, which we refer to as the Diffusion Classifier baseline in our experiments. We also further explore a single object variant of Diffusion Classifier for this compound setting, where we condition the generative model on individual concepts (e.g., “a photo of a dog”) and select the two classes with the highest score. Details of both methods are detailed in A.5. Since the Diffusion Classifier Variant uses individual concepts to fit multi-concept images, it has worse performance than Diffusion Classifier baseline.

---

> > ### Comment · Reviewer_7Q2x · 2025-04-04
> >
> > Thank you for your response. Several of my concerns have been addressed. However, in the zero-shot setting, instead of implementing specific methods directly into the approaches, a common practice for handling multi-concept or multi-condition scenarios is to use prompt weighting or tools like the Compel package. I'm curious whether adopting such methods would lead to worse performance.
> >
> > As mentioned above, I will maintain my current score.

---

> > > ### Author Response · Authors · 2025-04-06
> > >
> > > Thank you for engaging in the discussion and for your thoughtful question! As suggested by the reviewer, we have added an additional comparison to see whether prompt weighting could solve the zero-shot perception task, using the Compel package. Specifically, we applied prompt weighting to the compound prompts as follows:
> > >
> > > 1. “a photo of a cat++, a dog, and a rabbit”
> > >
> > > 2. “a photo of a cat, a dog++, and a rabbit”
> > >
> > > 3. “a photo of a cat, a dog, and a rabbit++”.
> > >
> > > The idea is that if the image contains specific concepts (e.g., a cat and a dog), then the prompts “a photo of a cat++, a dog, and a rabbit” and “a photo of a cat, a dog++, and a rabbit” are expected to result in lower denoising error (higher likelihood) than the prompt “a photo of a cat, a dog, and a rabbit++”. To determine what two objects present in the scene, we choose the two prompt weighted compound prompts that have the lowest denoising error. We report the zero-shot perception accuracy in the table below. It can be seen that using Compel results in worse performance compared to our proposed approach. This suggests that simple prompt weighting may not be sufficient for effective multi-concept inference in this setting.
> > >
> > > | Models      | Accuracy |
> > > | ----------- | ----------- |
> > > | Diffusion Classifer     | 63.8   |
> > > | Compel (per reviewer's suggestion)  | 35.2     |
> > > | Ours | 80.3       |

---

### Decision · Program_Chairs · 2025-05-01

**Decision:**

Accept (poster)

**Comment:**

This paper proposes an inverse generative modeling approach to understand attributes of a single image. After the compositional generative model is trained, the inverse generative modeling algorithm is used to search for condition parameters that maximize the log-likelihood from a single input image as the inference results. Some major concerns raised by the reviewers including:

1. Novelty: the proposed method is an extension of an existing approach (Reviewer 7Q2x, Reviewer 6fxB);

2. Complexity: the size of the search space will increase exponentially (Reviewer 6fxB); The computational complexity involved when enumerating concept combinations (Reviewer Tv9z);

3. Datasets: the custom small-animal dataset are too trivial (Reviewer 7Q2x); The datasets used in the experiments mainly contain simple concepts (Reviewer xCEx);

4. Practicality: It's no practical to scale (in its current state) to more complex, realistic real world scenes (Reviewer xCEx); How feasible would it be to extend this method to more "involved" types of compositions (Reviewer s7kA).

The authors tried to address most of the concerns and four reviewers are positive to this submission. After several rounds of discussion, Reviewer 6fxB is still not fully convinced and has major concerns on novelty and complexity/practicality:

> Reviewer 6fxB: I agree with Reviewer 7Q2x that this work is an extension of an existing approach. The idea of using generative models for classification is not unique to this paper—the Generative Classifier (GC) baseline already adopts a similar strategy. This paper extends that idea by using compositional generative models, but the current version does not adequately address the combinatorial explosion problem when scaling to a large number of concepts.

Despite the average positive rating, I agree with Reviewer 6fxB that the authors should clearly acknowledge the trade-off between computational efficiency and accuracy in the revision.